# The effect of a political crisis on performance of community forests and protected areas in Madagascar

Rachel A. Neugarten [1,2] ✉, Ranaivo A. Rasolofoson [3,4], Christopher B. Barrett [5,6], Ghislain Vieilledent [7] & Amanda D. Rodewald [1,2]

Understanding the effectiveness of conservation interventions during times of political instability is important given how much of the world's biodiversity is concentrated in politically fragile nations. Here, we investigate the effect of a political crisis on the relative performance of community managed forests versus protected areas in terms of reducing deforestation in Madagascar, a biodiversity hotspot. We use remotely sensed data and statistical matching within an event study design to isolate the effect of the crisis and post-crisis period on performance. Annual rates of deforestation accelerated at the end of the crisis and were higher in community forests than in protected areas. After controlling for differences in location and other confounding variables, we find no difference in performance during the crisis, but community-managed forests performed worse in post-crisis years. These findings suggest that, as a political crisis subsides and deforestation pressures intensify, community-based conservation may be less resilient than state protection.

Much of the world's biodiversity is concentrated in nations with fragile governance systems exposed to repeated political crises[1] that can threaten biodiversity[2] and its associated benefits to people. Given recurrent political instability and ongoing biodiversity declines in many nations, there is an urgent need to identify which conservation interventions are most resilient during times of crisis and post-crisis recovery. Yet, there are no published studies of the relative performance of different kinds of conservation interventions during a political crisis. Here, we investigate how a political crisis affects the relative performance of community-managed forests and protected areas administered by Madagascar National Parks (MNP).

Community forest management (CFM) and government-administered protected areas are among the most widespread conservation interventions around the globe. CFM has been promoted as an alternative to strict state-managed protected areas to avert deforestation while also supporting the rights and interests of local

people[3]. Identifying the conditions under which community-based or state management may be more effective at conserving biodiversity remains a key research question[4,5]. In more remote areas, local people may be better able to protect forests due to higher costs of centralized monitoring and enforcement[6]. Alternatively, state management may be more effective at conserving biodiversity if local institutions are weak or incentives for local conservation are insufficient[4].

From a conservation perspective, there is evidence that local communities can conserve vulnerable ecosystems better than the state under certain biophysical, economic, cultural, or sociopolitical conditions[3,7]. For example, community forests were found to be more effective than state-managed protected areas in terms of reducing deforestation in Peru[8]. Community forests were effective at reducing deforestation relative to a counterfactual in India[9] and Indonesia[10] and at reducing forest disturbance in Tanzania[11]. A systematic review found

[1]Department of Natural Resources and Environment, Cornell University, 226 Mann Drive, Ithaca, NY 14853, USA. [2]Cornell Lab of Ornithology, Cornell University, 159 Sapsucker Woods Rd, Ithaca, NY 14850, USA. [3]Duke Marine Lab, Nicholas School of the Environment, Duke University, 135 Duke Marine Lab Rd, Beaufort, NC 28516, USA. [4]School of the Environment, University of Toronto, 33 Willcocks Street, Suite 1016V, Toronto, ON M5S 3E8, Canada. [5]Charles H. Dyson School of Applied Economics and Management, Cornell University, Ithaca, NY 14853-7801, USA. [6]Jeb E. Brooks School of Public Policy, Cornell University, Ithaca, NY 14853-7801, USA. [7]AMAP, Université de Montpellier, CIRAD, CNRS, INRAE, IRD, Montpellier, France. ✉e-mail: rneugarten@wcs.org

that decentralized systems of forest management reduce deforestation, on average, but the effects are small[12].

Evaluating the performance of interventions requires eliminating rival explanations for observed outcomes[13]. For example, many protected areas are established in remote areas or in areas that are unsuitable for agriculture and therefore are unlikely to experience deforestation even in the absence of protection[14]. This makes it challenging to isolate the effects of different conservation interventions from other factors, such as remoteness. For example, multiple-use protected areas in Bolivia, Costa Rica, Indonesia, Thailand, and Brazil were found to be just as effective, or more effective, than strictly protected areas at avoiding deforestation[15,16]. This was because multiple-use areas are more likely to be located in areas with higher deforestation pressures, such as closer to roads and cities, where even modest reductions in deforestation were significant. Given differences in accessibility and other confounding factors, evaluating the relative performance of different interventions is challenging. This compromises our ability to test assumptions and design impactful conservation strategies.

A growing number of studies attempt to define what would have happened under a counterfactual scenario in an effort to isolate the causal effects of conservation interventions[8,15–17]. A systematic review of 68 such studies found that estimates of the effectiveness of

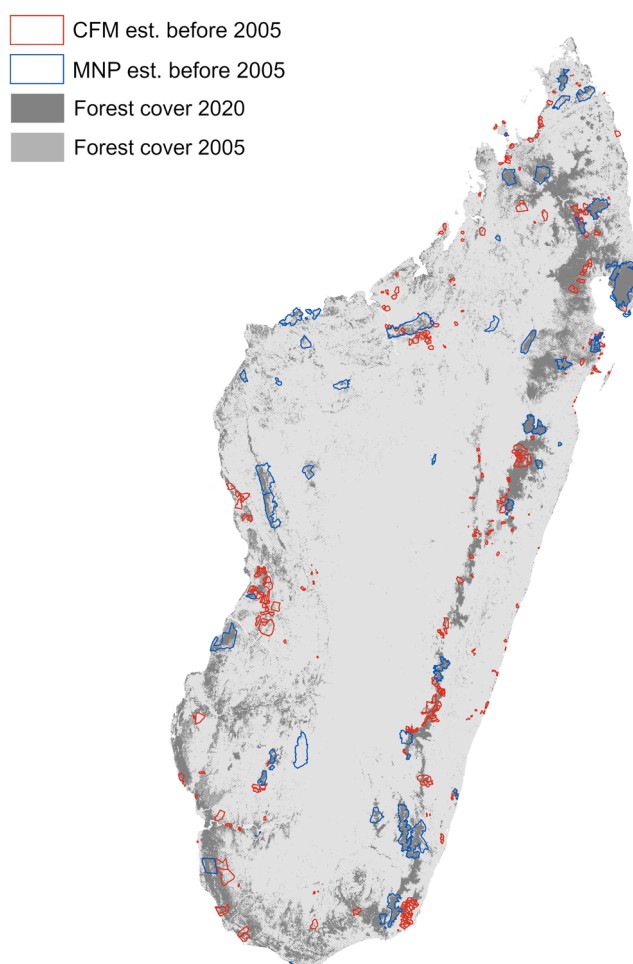

**Fig. 1 | Map of Madagascar showing Community Forest Management areas (CFM) established before 2005 (red outline), protected areas administered by Madagascar National Parks (MNP) established before 2005 (blue outline).** Forest cover 2020 (dark gray), forest cover loss 2005–2020 (medium gray), and all other land cover classes (grassland, shrubland, cropland, and urban) (light gray). For CFM and protected areas established after 2005 (see Fig. S1).

protected areas in terms of avoided deforestation were much smaller when counterfactual methods were used compared to traditional methods[18]. A second review of 82 counterfactual-based studies found that protected areas were only moderately effective at reducing deforestation, on average, since they are typically placed in areas with lower pressures[17]. Other interventions, such as decentralized forest management and Indigenous protected lands, had larger effects, but the number of studies using counterfactual methods was very small (three studies in each case). In Peru, for example, Indigenous territories and locally managed conservation concessions were more effective than state-managed protected areas in terms of avoided deforestation and degradation after controlling for confounding factors such as distance to roads and settlements[8].

There have been very few counterfactual-based studies that investigate the effectiveness of conservation interventions in times of crisis. The few examples we identified focused on armed conflict. In Nepal, local institutions were able to organize and cooperate to reduce forest fragmentation even during periods of violent conflict[19]. In Colombia, large protected areas were more effective at reducing deforestation during periods of conflict between the government and guerilla fighters[20]. In Sierra Leone, armed conflict was linked to lower rates of deforestation, but the performance of conservation interventions was not specifically analyzed[21]. Two studies from Rwanda found that armed conflict led to increased deforestation[22,23], but did not control for potential confounding factors such as location or climate-related variables.

Only a few counterfactual-based studies have assessed how sociopolitical context can influence conservation performance. Abman[24] showed that deforestation rates were higher in less democratic nations that failed to control corruption or protect property rights. In Indonesia, direct elections boosted the ability of protected areas to prevent deforestation, but not forest fragmentation or fire[25]. Elections were found to increase deforestation in Brazil[26] and increase forest fires in Madagascar[27]. Previous periods of political instability have been associated with increased deforestation in Madagascar[28].

Despite these advances, we know little about the relative effectiveness of different conservation interventions during and after a crisis. Here, we investigate how a political crisis affected the relative performance of community-managed forests compared to more traditional protected areas. Our evaluation focused on forests within Madagascar, a global biodiversity hotspot containing some of the most unique and threatened species on the planet[29]. An estimated eighty percent of Madagascar's people live under the extreme poverty rate of USD $2.15/day and 40% of children under the age of 5 suffer from stunting[30]. Due to these high levels of poverty and food insecurity, much of the island's population depends on natural resources, including forests, for their livelihoods. The country lost 44% of its natural forest cover over the period 1953–2014[31] due to logging for timber, charcoal production, and clearing for subsistence agriculture. A more recent analysis indicates that between 2000 and 2020, the country lost 4.85 million hectares, or 25% of its remaining tree cover[32].

Madagascar's government, often with support from international non-governmental organizations (NGOs), has attempted to slow deforestation through the creation of protected areas. As of 2020, the protected areas system included 110 sites encompassing 10.4% of Madagascar's land area (6.1 million hectares)[33]. From 1990 to 2010, Madagascar's protected areas were found to be effective at reducing deforestation, on average, but performance varied across time and space[34]. In northeastern Madagascar, for example, the establishment of new protected areas initially exacerbated ongoing deforestation but later reduced forest loss[35]. Protected areas in Madagascar are managed by different government agencies and NGOs. Here, we focused on 45 protected areas administered by Madagascar National Parks (MNP), an organization mandated by the government to manage protected areas (Fig. 1). MNP sites include some of Madagascar's oldest protected

areas, which are managed primarily for biodiversity conservation, and restrict most human activity other than recreation.

In the 1990s, the Madagascar government instituted legislation that allowed for the creation of Community Forest Management (CFM) contracts[36] (Fig. 1). Contracts are established between a local forest management group (often supported by a non-governmental organization), the federal forest department, and in some cases, the local government. The terms of CFM contracts vary, but they typically prohibit forest clearing for agriculture while allowing local use of renewable forest products for medicine, firewood, and food[37]. CFM contracts are established for an initial 3-year period, and if all parties agree that the site is being properly managed, the contract can be renewed for a subsequent 10-year term. The first CFM contract was established in 1999. By 2014, there were over 1000 CFM sites in Madagascar encompassing more than 3.1 million hectares, or 15% of the nation's natural forests[38]. Previous research found that CFM had no detectable impact on deforestation, on average, between 2000 and 2010, but contracts that prohibited commercial use of forest products did reduce deforestation[39].

Madagascar has experienced repeated political crises since its independence in 1960. The most recent and prolonged crisis took place from 2009 to 2014, initiated by a global spike in rice prices, a large, surreptitious land deal between the government and a South Korean company, and frustration over corruption and oppressive governance[40,41]. Social unrest and political pressure led then-President Marc Ravalomanana to flee the country while an opponent, Andry Rajoelina, took power. The international community condemned the takeover as unconstitutional and reduced or eliminated foreign aid and investment[42], causing a severe economic crisis. The crisis dragged on for years, with disastrous effects. In late 2013, democratic elections were held, and the crisis officially ended in January 2014.

The political and associated economic crisis also impacted Madagascar's forests and biodiversity. There was a spike in illegal logging of precious hardwoods such as rosewood[43,44]. Even within protected areas, illegal and extralegal logging took place as a result of the limited capacity of park staff and confusion caused by shifting regulations, in some cases with government permission or even cooperation[44-46]. Increased deforestation during the political crisis, both inside and outside of protected areas, alarmed the international conservation community, which was concerned about the potential extinction of Madagascar's unique wildlife, such as lemurs[2].

At the same time, the crisis exacerbated pressures on community forests. Combined with already high rates of poverty and food insecurity, the crisis drove local people to clear forests to plant staple crops and meet their basic needs. In northeastern Madagascar, for example, agricultural expansion into forests increased during the crisis period[35]. Political unrest can also result in deliberate forest burning as a form of protest. There is evidence of excess forest fires that coincide with Madagascar's 2013 and 2018 presidential elections, for example[27].

Given that Madagascar's forests faced concomitant pressure from government dysfunction, political protest, and economic stress, we explored the effect of the 2009 political crisis on the relative performance of community forest management areas (CFM) and protected areas administered by Madagascar National Parks (MNP) during and after the crisis. Performance was defined as the ability to reduce deforestation after controlling for differences in location and other potentially confounding factors. Our study built upon prior work that evaluated the overall effectiveness of protected areas 1990–2010[34] and CFM 2000–2010[39], though without reference to political crisis. As such, we had no clear a priori predictions of which type of area would perform better amid the crisis. To provide a sufficient pre-crisis baseline, we focused on 362 CFM sites established prior to 2005, as well as 45 protected areas administered by MNP, which were all also established prior to 2005 (Fig. 1).

To allow causal interpretation of our results, we used deforestation data derived from remote sensing[31,47] and a counterfactual approach implemented through a combination of statistical matching and an event study design. We combined two methodological approaches to control for factors that can confound the estimated relative performance of CFM and MNP. First, we used statistical matching[48] to identify forest areas within CFM and MNP that are similar across a range of observed biophysical and geographic confounding characteristics, such as remoteness and suitability for agriculture. This allowed us to make an apples-to-apples comparison of similar forested sites within CFM and MNP-protected areas, controlling for time-invariant confounding factors. Second, we conducted an event study analysis[49] to control for all relevant observed time-variant confounding factors, such as rice prices and climate variables. The event study also allowed us to control for any differences in deforestation trends in CFM and MNP in the pre-crisis period to isolate the effect of the crisis. Additionally, an event study design allowed us to examine the yearly variation in the effect of the crisis on the relative performance of CFM compared to MNP. We also explored the effects of spatial resolution on our results. Lastly, because the impacts of the crisis on CFM performance may vary as a function of contextual variables, we explored the moderating effects of contextual factors such as distance to cities, distance to roads, and population density. In our study, the event was the onset of the crisis, and our specific research question was, "What was the effect of the crisis and post-crisis period on the relative performance of CFM and MNP in terms of their ability to reduce deforestation?"

We found that during the crisis (2009–2013), CFM and MNP performed similarly poorly, meaning both experienced increasing deforestation. At the end of the crisis and for several subsequent years (2014–2017), annual rates of deforestation accelerated, particularly within CFM. During this post-crisis period, CFM performed significantly worse than matched MNP areas, even after controlling for time-invariant and time-variant confounding factors. Given the recurring political and economic crises taking place in many countries, our findings raise questions about the ability of both state- and community-managed conservation mechanisms to withstand such shocks. Community forests were especially vulnerable when deforestation pressures intensified, indicating that in Madagascar, such areas may require external support if they are to achieve forest and biodiversity conservation goals.

## Results
### Deforestation rates
Before, during, and after the crisis, annual deforestation rates were approximately three times greater in CFM than in protected areas administered by MNP (Table 1, Fig. 2). Notably, annual rates of deforestation accelerated immediately following the crisis, from 2014 to 2017, particularly in CFM (Fig. 2). To put these numbers in perspective, during the study period (2005–2020) forest cover declined by 16.5% nationally, from 9.7 million hectares (mha) in 2005 to 8.1 mha in 2020 (Table S1, Fig. S2). During the same period, forest cover within MNP declined by 6.5% (from 1.2 mha in 2005 to 1.1 mha in 2020) and forest cover in CFM areas declined by 20.9% (from 487,900 ha in 2005 to 385,700 ha in 2020).

Importantly, a simple comparison of deforestation rates does not control for confounding variables that influence the likelihood that a site is designated as a CFM or MNP and also affects forest loss. In Madagascar, previous work has found that confounding variables include distance from the nearest road, distance from the nearest village, distance from the nearest urban center, distance from forest edge, slope, elevation, and agricultural suitability because such factors influence the type of designation as well as forest cover outcomes[34,39]. Also, dynamic events such as climate extremes (droughts, floods, cyclones) and price fluctuations (such as global rice prices and price volatility)[50] could differentially affect deforestation in CFM and MNP,

**Table 1 | Average annual deforestation rates before, during, and after the political crisis within CFM established before 2005, protected areas administered by MNP, other protected and unprotected forest (which includes unprotected forests, CFM established after 2005, and protected areas established after 2005 and/or administered by agencies other than MNP), and total (which includes all categories)**

| Time period | CFM | MNP | Other protected and unprotected forest | Total |
|---|---|---|---|---|
| Forest cover in 2005 (sq km) | 4879 | 12,063 | 79,746 | 96,689 |
| Pre-crisis 2005–2009 (%) | 0.7 | 0.2 | 0.6 | 0.7 |
| Crisis 2010–2014 (%) | 1.1 | 0.4 | 0.9 | 1.0 |
| Post-crisis 2015–2020 (%) | 2.0 | 0.7 | 1.2 | 1.3 |

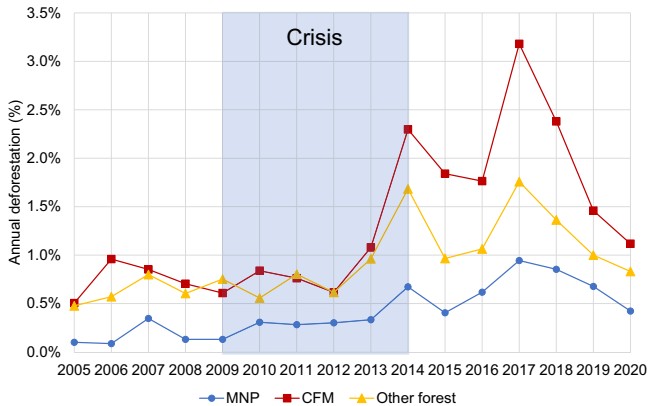

**Fig. 2 | Annual deforestation (as a percentage of 2000 forest cover) 2005–2020 in community forest management areas (CFM, red squares, protected areas administered by Madagascar National Parks (MNP, blue circles), and the rest of the country (other forest, yellow triangles).** Deforestation percentages are based on baseline forest cover in 2000, the earliest year for which there is consistent annual data. The crisis period (2009–2014) is shown in light blue shading.

which could bias results from a naïve comparison of deforestation outcomes. Such naïve comparisons fail to account for the fact that different types of sites experience very different levels of background deforestation pressures[18]. Therefore, we conducted statistical matching followed by an event study design to control both time-invariant confounding variables (such as remoteness) and time-variant confounding variables (such as climate).

### Match balance
Our units of analysis were 90 m grid cells that contained forest cover in the baseline year (2005). Before matching, the CFM forest grid cells, when compared to MNP forest grid cells, were more accessible (closer to cart tracks, roads, and villages) and had higher human population density, on average (Fig. 3). They were also closer to the forest edge as of 2005, lower in elevation, with lower slope, less annual precipitation, and located in more arid vegetation zones, on average. Because of these factors, forests within CFM would have had a higher probability of being deforested, on average, than forests within MNP, in the absence of effective protection. After matching, most of these differences were eliminated (Fig. 3). Distance to urban centers and suitability for rice cultivation were already very similar pre-matching (<0.1 standard deviations). In other words, after matching, matched forest grid cells in MNP were very similar to typical CFM forest grid cells and therefore provided a more useful apples-to-apples comparison. For maps illustrating differences in CFM and MNP sample points before and after matching (see Figs. S4 and S5). For results of alternative matching procedures (see Fig. S6).

### Event study
The event study results indicated that the political crisis affected the performance of CFM and MNP. During the first four years of the crisis,

CFM and MNP performed similarly poorly, meaning both experienced increasing deforestation (Fig. 4, Table 2, see also Table S2, Fig. S9). (Note that the event study design linearly controls for differences in pre-crisis trends (2005–2009) so 2010 is the first crisis year reported.) CFM performed significantly worse than matched MNP areas during the last year of the crisis and for several subsequent years (2014–2017) ($p < 0.05$). The difference in the effect of CFM relative to MNP on deforestation in the years 2014–2017 ranged from $1.7 \pm 1.4\%$ per year to $2.4 \pm 1.0\%$ per year. In other words, CFM had higher annual deforestation than MNP in those years, even after controlling for differences in location and other confounding variables. In the year immediately preceding the crisis (2008), CFM contained 475,333 ha of forest (Table S1). Thus CFM forests lost an estimated $8103 \pm 6435$ ha/year to $11,532 \pm 9508$ ha/year more tree cover than similar forests in MNP. This is equivalent to a total of $36,483 \pm 28,775$ ha ($51,047 \pm 40,228$ soccer fields) for the 2014–2017 period. From 2018 to 2020, CFM continued to perform worse than MNP, but the difference is no longer statistically significant.

We clustered standard errors at the site level to address potential spatial autocorrelation between observations within the same site[51]. We also tested multilevel clustering of standard errors at the site and region level. Multilevel clustering did not affect our point estimates but rendered the observed differences in the years 2014–2017 marginally significant ($p < 0.1$ instead of $p < 0.05$).

### Tests of heterogeneity of impacts and spatial resolution
Our results were consistent for the sub-set of CFM for which the contracts were renewed, which we used as an indicator of the level of CFM implementation on the ground. That is, we found no significant difference in renewed CFM and matched areas within MNP during the crisis years, and renewed CFM performed worse in the years immediately following the crisis (Table S3). Our results were also robust to the spatial resolution of the input data at the two resolutions we tested (90 and 270 m). At a coarser spatial resolution, the observed difference in performance in the post-crisis years had greater statistical significance ($p < 0.01$) (Table S4).

We found that CFM further from urban centers performed better than those closer to cities in the post-crisis period, and the difference was statistically significant in 2015, 2016, and 2018 (Fig. S10, Table S5). Thus, distance from urban centers, a measure of remoteness, appeared to influence the performance of CFM. Even in remote areas, however, CFM was less effective at reducing deforestation than similarly remote MNP. We explored potential heterogeneity of effects using other variables, including distance from roads, distance from villages, population density, level of development, and security, but found no consistent or significant effect of any of these variables (Tables S6–S8).

### Discussion
A disproportionate share of the world's biodiversity is concentrated in nations that are highly vulnerable to political and economic shocks, yet few studies have examined how conservation interventions perform during and after crises. We found that, despite conservation efforts that sought to protect forests during Madagascar's recent political

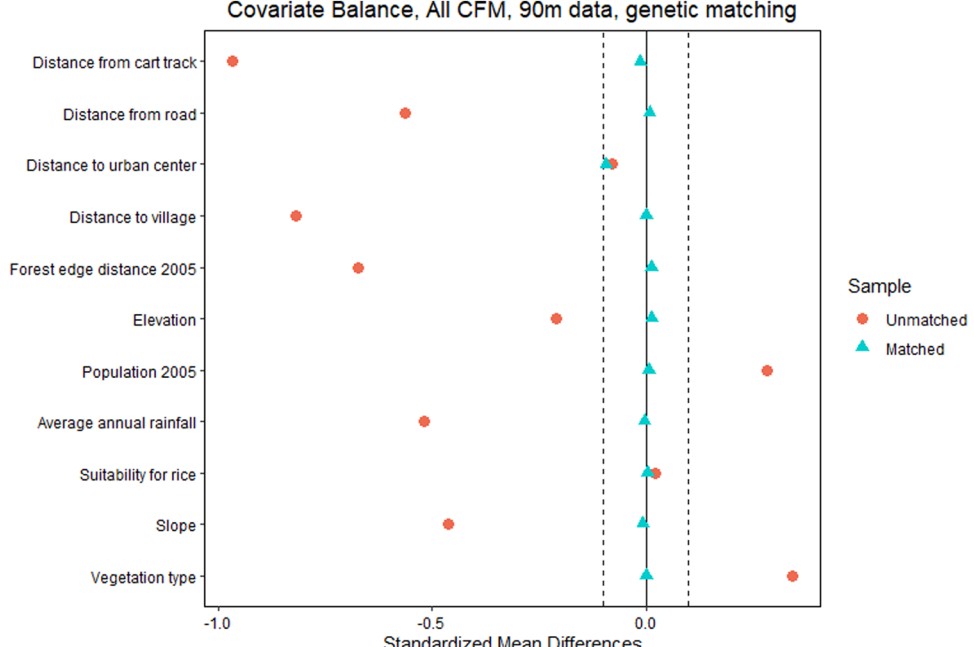

**Fig. 3 | Covariate balance between community forest management areas (CFM) and protected areas administered by Madagascar national parks (MNP) forest grid cells, before (red circles) and after (blue triangles) genetic matching.** Black dotted line indicates a standardized mean difference of 0.1 standard deviations. Vegetation zone codes: 1 = Eastern humid forest; 2 = Western deciduous forest; and 3 = Southern deciduous spiny forest, so higher values indicate drier forest types.

crisis, annual rates of deforestation accelerated at the end of the crisis —a phenomenon that, to our knowledge, has not been reported previously. Understanding the cause of this post-crisis increase in deforestation is beyond the scope of this analysis, but we can provide some theories that could be explored in future work. One possibility is that we detected a lagged response to events that occurred during the crisis. Funding for conservation declined precipitously during the crisis[2,52], and it took several years for financial support to be restored to pre-crisis levels. Weak governance and increased corruption during the crisis[2,46] might have had lingering effects or become more severe in the post-crisis period.

The years in which we observed an increase in deforestation also roughly coincided with Madagascar's post-crisis presidential elections (December 2013, November 2018). Disputed elections can trigger social unrest, and even peaceful transitions can usher in forest policy change. In Brazil, for example, election cycles were found to trigger deforestation[26]. The 2013 and 2018 Madagascar presidential elections were associated with excess forest fires, which may indicate burning as a form of political protest[27]. The relationship between deforestation and political instability can be hard to untangle, however. Logging precious hardwoods provided a source of cash for a wealthy elite during the crisis[46] and may have continued or increased in the post-crisis era. At least one study speculates that the wealth created by exploiting forest resources in Madagascar can be politically destabilizing[46], indicating that deforestation could contribute to a crisis rather than the other way around.

Another possible explanation for the post-crisis deforestation spike is that the return to political stability in the post-crisis period might have initiated a change in forestry policy or triggered an increase in economic activity, putting even more pressure on forests. We found no evidence of a formal change in forestry policy during the post-crisis era, however, per-capita GDP did not increase substantially during 2014–2017[30]. What drove the observed post-crisis deforestation spike therefore requires further study.

While overall deforestation dynamics during and after the crisis are important, our focus here was on conservation performance. Given

the recurring political and economic crises taking place in many countries, our findings raise questions about the ability of both state- and community-managed conservation mechanisms to withstand such shocks. Given how differently CFM and MNP are designated and managed, the finding that there was no significant difference in their performance during the crisis was unexpected. It seems that neither form of forest protection was durable, likely due to a lack of capacity and resources to enforce rules. Even at the best of times, protected area managers in Madagascar struggle to implement regulations due to limited budgets and lack of political support[52]. During the crisis, lack of capacity and legal authority prevented park staff from controlling illegal logging or agricultural expansion in national parks[45]. Communities were probably similarly ill-equipped to protect their forests during the crisis years.

At the end of the crisis, community-managed forests performed significantly worse than protected areas administered by MNP, with annual deforestation rates 1.7–2.4 times higher, even after controlling for differences in location and other confounding factors. The years in which we observed a difference in performance (2014–2017) correspond to the overall increase in deforestation across the country. In other words, when deforestation pressures intensified, community-managed forests proved more vulnerable than MNP-managed forests. Poor performance of CFM was also described by Rasolofoson et al.[39]. This is sobering, as community-based conservation is often promoted as an effective and equitable alternative to traditional, government-run protected areas. It is likely that such differences reflect the lack of capacity and resources of communities to protect forests. CFM receives no centralized financial support in Madagascar, which may have made them particularly vulnerable to the loss of conservation funding during and after the crisis.

After 2017, CFM continued to perform worse than MNP, but the differences were no longer statistically significant. That would be consistent with differential rates of recovery among CFM, some of which began to converge back towards MNP within a few years of the crisis end, others of which continued to lag far behind. This makes sense, given the heterogeneous management of CFM and relatively

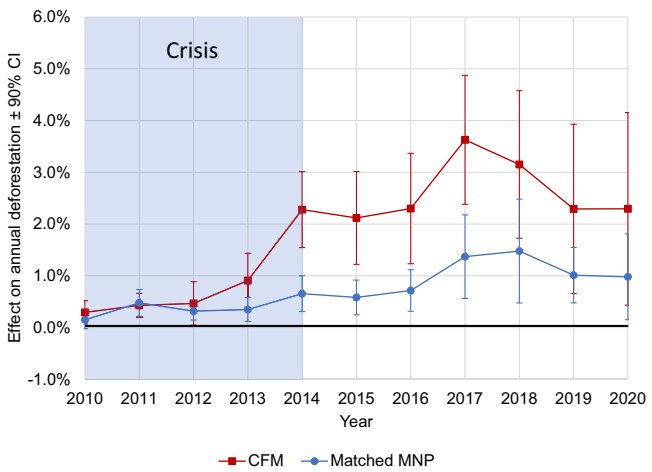

**Fig. 4 | Event study results. Effects of the political crisis on the performance of community forest management areas (CFM, red squares) and protected areas administered by Madagascar national parks (MNP, blue circles) in terms of annual deforestation, after matching and controlling for time-variant covariates.** Estimates greater than zero indicate more deforestation (poor performance). *Y*-axis values represent the estimated effect size of the political crisis on annual deforestation (mean percent tree cover loss per year) from an ordinary least-squares regression using an event study design with fixed effects for forest grid cells, as described in Table 2. Error bars indicate 90% two-sided confidence intervals. Our sample consists of 11,626 observations within CFM and 4244 unique observations within MNP (matched observations). The difference between the red and the blue points each year indicates differential effects of the crisis on CFM relative to matched MNP areas. The event study analysis controls for trends in the pre-crisis period (2005-2009), so the first data point represents the first crisis year (2010).

**Table 2 | Results of event study model**

| Variable | Estimate | Standard error | Statistic | *p*-value |
|---|---|---|---|---|
| Year | −7.79E−04 | 2.79E−04 | −2.796 | 0.005** |
| 2010 | 2.29E−03 | 1.19E−03 | 1.92 | 0.056. |
| 2011 | 5.56E−03 | 1.77E−03 | 3.138 | 0.002** |
| 2012 | 3.97E−03 | 1.29E−03 | 3.079 | 0.002** |
| 2013 | 4.28E−03 | 1.65E−03 | 2.588 | 0.01* |
| 2014 | 7.34E−03 | 2.33E−03 | 3.149 | 0.002** |
| 2015 | 6.61E−03 | 2.30E−03 | 2.871 | 0.004** |
| 2016 | 7.95E−03 | 2.68E−03 | 2.967 | 0.003** |
| 2017 | 1.45E−02 | 5.12E−03 | 2.827 | 0.005** |
| 2018 | 1.56E−02 | 6.28E−03 | 2.48 | 0.014* |
| 2019 | 1.09E−02 | 3.49E−03 | 3.128 | 0.002** |
| 2020 | 1.06E−02 | 5.27E−03 | 2.009 | 0.045* |
| Distance from forest edge | −5.32E−05 | 8.45E−06 | −6.296 | 0*** |
| Population density | −5.21E−06 | 4.07E−05 | −0.128 | 0.898 |
| Average rice price | −5.41E−09 | 3.57E−09 | −1.514 | 0.131 |
| Standard deviation in rice price | 2.16E−08 | 1.34E−08 | 1.615 | 0.107 |
| Drought severity (−) | −3.46E−06 | 1.94E−06 | −1.787 | 0.075. |
| Maximum precipitation | 5.20E−06 | 2.90E−06 | 1.792 | 0.074. |
| Maximum temperature | −6.50E−05 | 8.04E−05 | −0.808 | 0.42 |
| Maximum wind speed | −6.46E−06 | 1.22E−05 | −0.53 | 0.597 |
| CFM:Year | −1.69E−03 | 1.15E−03 | −1.477 | 0.141 |
| CFM:2010 | 3.13E−03 | 2.53E−03 | 1.236 | 0.217 |
| CFM:2011 | 1.22E−03 | 2.99E−03 | 0.407 | 0.684 |
| CFM:2012 | 3.18E−03 | 3.56E−03 | 0.895 | 0.372 |
| CFM:2013 | 7.28E−03 | 4.59E−03 | 1.587 | 0.113 |
| CFM:2014 | 1.79E−02 | 5.94E−03 | 3.018 | 0.003** |
| CFM:2015 | 1.70E−02 | 6.91E−03 | 2.468 | 0.014* |
| CFM:2016 | 1.75E−02 | 7.84E−03 | 2.237 | 0.026* |
| CFM:2017 | 2.43E−02 | 1.02E−02 | 2.377 | 0.018* |
| CFM:2018 | 1.84E−02 | 1.15E−02 | 1.6 | 0.11 |
| CFM:2019 | 1.45E−02 | 1.15E−02 | 1.256 | 0.21 |
| CFM:2020 | 1.48E−02 | 1.33E−02 | 1.119 | 0.264 |

The event study is an ordinary least squares regression with the following variables: Dependent variable: annual deforestation (percentage). Treatment variable: CFM (takes value 1 for CFM, 0 for MNP). Year: 2005–2020. Years post-crisis: 2010–2020. Time-variant covariates (all are annual): distance from forest edge (meters), population density (people per square km), average annual rice price converted to Madagascar currency (Ariary), standard deviation of rice price converted to Madagascar currency (Ariary), drought severity (Palmer Drought Severity Index, more extreme negative numbers indicate more severe drought), maximum precipitation (mm), maximum temperature (°C), and maximum windspeed (m/s), an indicator of cyclone severity. Individual fixed effects were included for each forest grid cell (*n* = 23,252), and standard errors clustered at the level of sites (where each site is a unique CFM (*n* = 362) or MNP (*n* = 45)). The coefficients of interest are the interactions between CFM and Years post-crisis. Significance codes: '***' 0.001; '**' 0.01; '*' 0.05; '.' 0.1. Root mean squared error: 0.065427, adjusted *R*-squared: 0.024156, within *R*-squared: 0.017623.

more homogeneous management of MNP-protected sites. We found no significant effect of climate-related variables nor other time-variant controls. Thus, it is unlikely that the difference in performance was driven by events such as a drought, a cyclone, or rice price fluctuations.

The estimated difference in performance between CFM and MNP in the post-crisis years may seem small (1.7–2.4%/year), but given that CFM contained nearly half a million hectares of forest at the start of the crisis, such a difference is ecologically meaningful. Also, our estimated effect sizes are in line with other studies that use quasi-experimental methods to estimate the effects of protected areas on avoided deforestation, which are typically under 5%, and often under 1%[18].

In areas further from cities, CFM performed better but still not as well as MNP. In more remote areas, community-based conservation may be more effective due to lower costs of monitoring and enforcement. Remote areas are also somewhat isolated from economic pressures, which may incentivize forest clearing. Nonetheless, remote MNP-administered sites still withstood deforestation pressures better, so our results indicate that the type of designation a site receives is an important determinant of performance. Previous research from a subset of four sites indicates that CFM contracts that prohibited commercial use performed better than CFMs that allowed such uses[39]. For the vast majority of CFM, however, there is a lack of data on specific contract terms or their implementation. Our analysis therefore focuses on *de jure* designation, as currently, there is no national-scale data on the de facto management of CFM or MNP. Future research on how CFM and MNP are managed in practice would expand our understanding of how they interact with shocks such as political crises.

In addition to protecting forests, CFM was designed to benefit local communities. While increased deforestation is undesirable from a conservation perspective, income from clearing forests may have provided a safety net to local people during the post-crisis period. Previous research indicated that CFM provides economic benefits to

households in close proximity to forests[38]. Alternatively, poor CFM performance following the crisis might have been driven by incursions to the forest by outside actors or resource capture by local elites. Past work indicates that, in some communities, CFM contracts were unable to prevent illegal logging by companies or migrant groups[37]. In such cases, strengthening tenure security might contribute to improved outcomes, both for forests and for local people[53,54]. Tenure security is likely to be impacted by a political crisis, however, given that the state may no longer enforce land tenure claims.

Our study design linearly controls for differences in pre-crisis trends, all time-invariant confounding factors, and many time-variant confounders (see the "Methods" section). Nonetheless, we cannot completely rule out the effect of unobserved time-variant confounders. Remote sensing of forest cover has made considerable advances in the past two decades but still has limitations, particularly for detecting selective logging. A comparison of global and locally calibrated forest cover change datasets in Masoala National Park in northeastern Madagascar indicates that both datasets performed reasonably well in detecting small slash-and-burn agriculture, but neither did a good job of detecting selective logging[55]. Limitations in remotely sensed data can bias estimates of causal impacts of conservation, particularly in areas with high cloud cover and steep slopes[56], which may affect our results in the humid eastern highland forests of Madagascar. Because we used an event study design that compared forest areas to themselves over time and exact matching within each vegetation zone, however, we believe any such effects would not change our qualitative conclusions.

Our analysis was somewhat complicated by issues of overlapping designation. In some cases, sites that were designated as protected areas by the government are partially or entirely managed by local communities. The portions managed by local communities are included in our analysis as CFM. Overlapping areas where management of the site is unclear were excluded from our analysis (see, for example, Figs. S4 and S5).

Our results compare CFM to similar (matched) areas within MNP and do not reflect the overall performance of protected areas administered by MNP. CFM forests are more accessible (closer to roads and villages), lower in elevation, with lower slope, closer to forest edges, and are otherwise different from MNP. Our matched dataset, therefore only includes forested areas of MNP that are similarly accessible, lower elevation, closer to the forest edge, and otherwise comparable to CFM forests. Our results are thus representative of the performance of all CFM forests (in the event of a crisis) relative to similar forests within MNP but are not representative of MNP performance as a whole. Also, while site-level estimates of CFM conservation performance would be desirable, due to the small number of MNP (45) relative to CFM (362), and systematic differences between MNP and CFM, it is not feasible to find good matches at the site level. Hence, we identified forest grid cells within CFM and MNP that had similar characteristics for our analysis.

Conservation impact evaluations often use a difference-in-differences (DiD) design[39,57–59]. Here, we use an event study design as it has two advantages over a DiD: it linearly controls for differences in pre-crisis trends, and it has the flexibility to detect the changing impacts of the crisis over time. A DiD analysis would only estimate the average post-crisis impacts and, thus, would impose the strong assumption that all post-crisis impacts were equivalent over time[60]. These advantages render our design more robust to the identification assumption of parallel deforestation trends between CFM and MNP in the absence of the crisis (although we also found evidence of parallel trends in the pre-crisis period, see Supplemental Materials S1, Table S9, Fig. S11). Identification assumptions are not directly testable, however[13]. In addition to linearly controlling for differences in pre-crisis trends, our study design controls for all time-invariant confounding factors and a number of time-variant confounders. Our study design also controls for any time-variant factors that influence deforestation outcomes equally in CFM and MNP. Nonetheless, we cannot directly observe the counterfactual (what would have happened in the absence of the crisis). Therefore, we cannot completely rule out the effect of unobserved time-variant confounders. Furthermore, by clustering standard errors at the site level, we have addressed possible spatial autocorrelation between observations within the same site, but this does not eliminate the possibility of spatial autocorrelation among observations at different (nearby) sites[51].

Future work would benefit from improved data availability, including more detailed information on CFM contract terms, how CFM is managed in practice, and how CFM and MNP performance is influenced by land tenure insecurity. Finally, our results represent only one country and a single crisis, additional research on the performance of different kinds of conservation interventions during times of political instability is needed.

While recognizing the many challenges associated with isolating causal effects of conservation interventions, our research indicates that efforts to protect forests in Madagascar, especially community-based efforts, were vulnerable to the crisis and post-crisis dynamics. As such, improving the resilience of forest protection mechanisms to political and economic shocks is needed to avoid losing tropical forests during and after such crises. Recent research from eight countries, including Madagascar, indicates that social cohesion, recognition of community rights, and support of national authorities are critical for successful community forest conservation, especially when faced with threats from economically and politically powerful external actors[7]. A separate study of 643 CFM from 51 countries found that successful ecological and socioeconomic outcomes were more likely for forests with local tenure rights, co-management seed approaches, and smaller user groups[3]. Lessons from these and similar studies suggest potential pathways for improving CFM performance during and after crises, such as strengthening social cohesion, reinforcing local tenure and use rights, and increasing levels of support from government and non-governmental conservation organizations.

## Methods
### Deforestation
The outcome variable of interest is deforestation in a given year[31]. Deforestation is widely used to measure the environmental impacts of conservation interventions (see Ribas et al.[18] for a recent review). We calculated deforestation as the change in forest cover (as a percentage of each grid cell, 0–100%) in a given year ($t$), where positive values indicate forest loss:

$$\text{Deforestation} = \text{forest cover}_{t-1} - \text{forest cover}_t \qquad (1)$$

To calculate deforestation, we started with forest cover data for Madagascar for the year 2000[31], the first year in which high-resolution (30 m), standardized annual forest cover was available. The 2000 forest cover product was produced based on satellite imagery (Landsat TM and ETM+)[61] combined with a 2000 tree cover percentage map[47] to fill gaps due to the presence of clouds[31]. We combined 2000 forest cover with global 30 m tree cover change estimates[47] to generate an annual time series from 2000 to 2020 (Fig. S2). The resulting deforestation data takes values 1 for deforestation and 0 for no deforestation in each 30 m grid cell. To match the spatial resolution of our other covariate data (90 m), we aggregated the deforestation data to two coarser spatial resolutions (90 and 270 m), resulting in a percentage of each 90 or 270 m grid cell that was deforested in each year (0–100%). The aggregation step was also done to convert a binary outcome variable to a quasi-continuous variable which allowed us to perform subsequent statistical analyses. We used annual deforestation (change within a single year) as the outcome variable as it is stationary (that is, it does not get larger every year), and stationarity is assumed for the statistical procedure we used in our time series analysis (event study). We did not include forest regrowth in our analysis because there is little evidence of natural forest regeneration in Madagascar due to burning, soil erosion, and reduced seed bank following clearing; and because the only available data on tree gain[47] includes plantations, not natural forest regrowth[31]. For this analysis, we focus on a pre-crisis baseline (2005–2009), a crisis period (2010–2014), and a post-crisis period (2015-2020).

## Sampling

To establish forest trends during a baseline (pre-crisis, 2005–2009) period, we focus on 362 CFM sites established prior to 2005, and 45 protected areas administered by MNP also established prior to 2005. We use 2005 as the baseline year as it provides a sufficient pre-crisis baseline. The first seven CFM contracts were signed in 1999. After that, the number of CFM sites established rose each year; in 2005, for example, 111 CFM were established, bringing the total to 362. For comparison sites, we focused on protected areas administered by MNP, and excluded sites managed by other agencies or NGOs, to evaluate a consistent form of protection. Sample and comparison points were generated using the "create spatially balanced points" tool in ArcMap[62], which generates a random sample of points that roughly represent the same proportion of the total study area[63]. Our CFM sample consisted of 12,000 points within CFM areas with more than 0% forest cover as of 2005 (Fig. 5). Initially, a larger number of potential MNP comparison points was required to identify good matches (that is, MNP forest grid cells that are similar, on average, to CFM forest grid cells). We, therefore, initially generated a pool of 36,000 potential sample points within MNP to use in the matching step described below. For each sample and comparison point, we calculated 90 m raster values representing the outcome variable of interest (annual deforestation 2005–2020, Fig. S2), time-invariant covariates used for matching (Table 3, Fig. S3), and time-variant covariates used in the event study analysis (Table 4, Fig. S7) using the terra R package[64].

The degree to which CFM contract terms are implemented in practice likely varies among sites. Unfortunately, consistent data on implementation is not available for the majority of CFM sites. As mentioned above, after an initial 3-year period, if all parties agree that the CFM is being properly managed, the contract can be renewed for an additional 10-year period[37]. Therefore, we used contract renewal as an indicator that the CFM contracts were being implemented on the ground and were not agreements on paper only. We repeated our sampling within a sub-set of CFM contracts that were renewed. This resulted in a separate set of 12,000 sample points from renewed CFM, which were analyzed separately in subsequent steps (matching and event study analysis). We refer to these separate sets of sample points and results as "all CFM" and "renewed CFM." We analyzed spatial data using the "terra" package in R[64], QGIS[65], and ArcMap[62]. To test for the effect of spatial resolution on our results[66], we repeated all analyses at two spatial resolutions: 90 and 270 m. All data was projected in WGS_1984_UTM_Zone_38S.

## Statistical matching

The goal of statistical matching is to identify and control for observed confounding factors[48,67,68]. In our case, these include variables that influenced the likelihood that a site was designated as CFM or MNP and also influenced the probability of deforestation. Due to their difference in number, size, and other characteristics, it is impossible to find good site-level matches for individual CFM or MNP sites. Thus, we focus on 90 m (and 270 m) forest grid cells as the unit of analysis. The matching step allowed us to identify a matched sample which included a treatment group (CFM forest grid cells) and a comparison group (matched MNP forest grid cells that were statistically similar to the CFM grid cells). This allowed us to perform an apples-to-apples comparison.

As variables for matching, we included data on factors that have been shown to influence both the probability that a site is designated as a CFM or MNP and also affect the likelihood of deforestation (Table 3)[39]. Because protected areas administered by MNP include the oldest protected areas in Madagascar and were established primarily to shield biodiversity from human pressure. MNP are therefore, larger and more contiguous and located in more remote, higher elevation areas with fewer competing land uses[69]. CFM is a newer designation and is intended for multiple uses. CFM therefore tend to be smaller and located in areas with higher human pressures[39]. Thus confounding

factors include suitability for rice agriculture[70], elevation and slope[71], average annual precipitation 1970–2000[72], distance to forest edge in 2005 (calculated for this analysis), distance from roads, cart tracks, villages, and urban centers[39], population density in 2005[73], and vegetation zones[39] as these variables influence both the type of designation as well as deforestation outcomes. Because the goal of matching is to identify variables that might have influenced the original probability that a site was designated as CFM or MNP, all variables used in matching were time-invariant (such as elevation and slope) or represented the first year in the study period (for example, population density as of 2005), or earlier time periods (such as historic precipitation averages).

We conducted 1:1 genetic matching with replacement. See Figs. S4 and S5 for examples of sample points before and after genetic matching. Genetic matching is not a unique matching method, rather, it identifies a method (such as propensity score matching or matching based on Mahalanobis distance) that optimizes covariate balance[74]. We used the MatchIt package in R[75] and covariate balance was assessed using the cobalt R package[76]. We also tested two alternative matching methods, propensity score matching, and Mahalanobis distance matching, but genetic matching led to better balance in the covariates (Fig. S6).

Due to differences in biophysical and socioeconomic characteristics in different regions of the country, we expected the political crisis to have different impacts. Therefore, we conducted exact matching within similar vegetation zones (eastern humid forests, western dry deciduous forests, and southern dry spiny forests). Matched MNP points are not necessarily matched to points in neighboring CFM, however. The matching procedure resulted in the original set of 12,000 CFM points and 12,000 matched MNP points. In some cases, CFM and MNP boundaries overlap due to overlapping designations or ongoing negotiations between MNP and local communities. Sample and comparison points in overlapping areas were excluded from the analysis, resulting in a final set of 11,626 CFM and 11,626 matched MNP points (Figs. S4 and S5). We performed matching with replacement, so not all the matched MNP points were unique; that is, the same MNP sample point could have been matched with multiple CFM sample points. Therefore, the matched MNP sample consisted of 4244 unique points. We addressed this pseudoreplication in our matched datasets by clustering standard errors at the site level. Matching was performed separately for the sample points from the renewed CFM sites, resulting in a second matched dataset consisting of 11,886 renewed CFM and 11,886 matched MNP points. (Due to matching with replacement, the matched MNP sample consisted of 3155 unique points.)

We considered comparing CFM and MNP performance to the unprotected forest. However, by the time of the crisis and the post-crisis period, there was very little forest that was not under some kind of designation due to the expansion of Madagascar's protected area and CFM network after 2005 (Fig. S1). Further, the forest that remained unprotected would not serve as a good match for CFM or MNP forest due to differences in location and other characteristics.

## Event study

After matching, we conducted an event study analysis to investigate the effect of the political crisis (the event), i.e. compare CFM and MNP performance before, during, and after the political crisis. The goal of the event study analysis is to control for all time-invariant and observed time-variant confounding factors that may influence deforestation outcomes[77] differently in CFM and MNP. We also performed a two-period difference-in-differences (DiD) analysis, comparing the pre-crisis period (2005–2009) and the crisis period (2010–2014) (see Supplemental Materials S2, Table S10). We note that our event study model is a more general form of a DiD model that provides two advantages relative to DiD. It allows us to control for any differences in deforestation trends in CFM and MNP in the pre-crisis period. And it

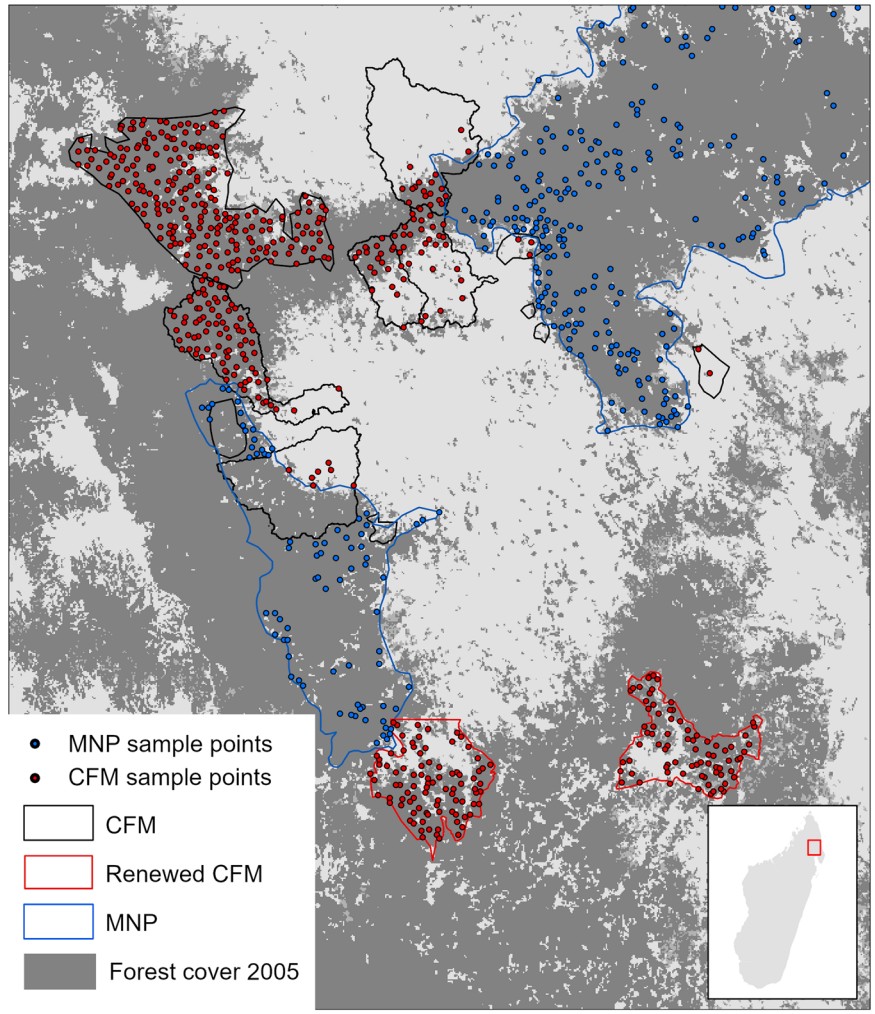

**Fig. 5 | Example of sampled areas in northeastern Madagascar including CFM (black and red outlines), the sub-set of CFM that were renewed (red outline), and MNP (blue outline).** Overlapping CFM and MNP areas were excluded from the analysis. Map shows randomly sampled points from CFM (red points) and matched forest grid cells within MNP (blue points). Forest cover in 2005 is shown in green. The MNP shown here includes Marojejy National Park (upper right) and Anjanaharibe-Sud National Park (lower left).

**Table 3 | Baseline characteristics that are likely to affect both assignments to CFM vs MNP and rate of deforestation, used as covariates in statistical matching**

| Covariate | Units | Spatial resolution | Source |
|---|---|---|---|
| Suitability for irrigated rice | 0 (unsuitable) or 1 (suitable) | 90 m | Ramaharitra Tondrasoa 2012[70] |
| Elevation | m | 90 m | Shuttle Radar Topography Mission (SRTM) Digital Elevation Model[71] |
| Slope | Percent | 90 m | SRTM[71] |
| Annual average precipitation (1970-2000) | mm/year | 90 m | WorldClim 2.1[72] |
| Distance to forest edge (2005) | m | 90 m | Vieilledent et al. (2018)[31] |
| Distance to a village (2005) | m | 90 m | Rasolofoson et al. (2015)[39] |
| Distance to an urban center (2005) | m | 90 m | Rasolofoson et al. (2015)[39] |
| Distance to a road (2005) | m | 90 m | Rasolofoson et al. (2015)[39] |
| Distance to a cart track (2005) | m | 90 m | Rasolofoson et al. (2015)[39] |
| Population density (2005) | People/km² | 90 m | WorldPop (2018)[73] |
| Vegetation type | 1 = Eastern humid forest; 2 = Western deciduous forest; 3 = Southern deciduous spiny forest | NA (polygons) | Rasolofoson et al. (2015)[39] |

**Table 4 | Time-variant variables expected to differentially affect deforestation within CFM and MNP, included as covariates in the event study analysis**

| Covariate (annual) | Units | Spatial resolution | Source |
|---|---|---|---|
| Distance to forest edge | m | 90 m | Vielledent et al. (2018)[31] updated to 2020; annual distance to forest edge calculated using Google Earth Engine[80] |
| Population density | People/km$^2$ | 1 km | WorldPop[73] |
| Maximum accumulated precipitation | mm | 5 km | TerraClimate[81] |
| Maximum temperature | °C | 5 km | TerraClimate[81] |
| Drought severity | Palmer drought severity index | 5 km | TerraClimate[81] |
| Maximum wind speed | m/s | 5 km | TerraClimate[81] |
| Average rice price | Madagascar Ariary | NA | World Bank[82] |
| Standard deviation of rice price | Madagascar Ariary | NA | World Bank[82] |
| *Time-invariant covariates used as interaction terms* | | | |
| Distance to a road | m | 90 m | Rasolofoson et al. (2015)[39] |
| Distance to a village | m | 90 m | Rasolofoson et al. (2015)[39] |
| Distance to an urban center | m | 90 m | Rasolofoson et al. (2015)[39] |
| Population density in 2005 | People/km$^2$ | 1 km | WorldPop[73] |
| Development level (index of material assets) in 2007 | 0 (below median) or 1 (above median) | Fonkontany (administrative unit) | Wu Yang, Conservation International, based on Communes Database[83] |
| Security conditions and risk of theft of property in 2007 | 0 (below median) or 1 (above median) | Fonkontany | Wu Yang, Conservation International, based on Communes Database[83] |

Interaction terms used to explore heterogeneity of impacts (see Supplemental Materials).

allows us to explore the yearly variation of the relative performance of CFM and protected areas (instead of only two time periods, pre- and post-crisis, in the case of a DiD). Thus, we focus on the event study here; results of the DiD are included in the Supplemental Materials. In both the DiD and event study analysis, we controlled for annual rice prices and rice price volatility[50], annual climate variables such as maximum precipitation, drought severity, and maximum wind speed (an indicator of cyclones)[27], annual population density[39], and annual distance to forest edge (Table 4, Fig. S7). All variables were spatially explicit (Fig S7) and were included on an annual basis for the years 2005–2020. We also controlled for each year so as to capture any other time-variant factors that could influence deforestation outcomes between years, but that would be common to all forest grid cells in the study. In addition to their potential for influencing deforestation outcomes, the selection of covariates was also influenced by data availability, as our study design requires data that are both temporally and spatially comprehensive (that is, available annually from 2005 to 2020 and for both CFM and MNP).

If deforestation in CFM was significantly different than deforestation in matched areas within MNP during the political crisis, after controlling for time-variant factors (such as rice prices or drought severity), we could attribute the difference to the interacting effect of the treatment (CFM vs. MNP designation) and the political crisis. For the event study analysis, we used an OLS regression with interaction terms representing the year (2005–2020), years post-crisis, and fixed effects for each spatial unit, using the fixest package in R[78]. Our event study model takes the form:

$$Y_{it} = \beta_1 CFM_i + \tau_1 year_t + \tau_2 year_t CFM_i + \gamma YearsPostCrisis_t + \delta CFM_i YearsPostCrisis_t + \psi X_{it} + \mu_i + \varepsilon_{it} \quad (2)$$

where $Y_{it}$ is forest cover loss (percentage) in each forest grid cell $i$ in year $t$ ($t = 2005$–$2020$); CFM = 1 if the grid cell falls within a CFM and CFM = 0 if it falls within an MNP; YearsPostCrisis = 0 for the crisis years (2005–2009) and then 2010, 2011, and so on; $X_{it}$ is a vector of time-variant controls (Table 4); and $\varepsilon_{it}$ is a random error term. We included individual fixed effects for each forest grid cell ($\mu_i$), and clustered standard errors at the site level (where each site is a unique CFM or MNP).

The unit of analysis was forest grid cells. To explore the potential effect of spatial resolution on our results[66], the analysis was conducted at two different spatial resolutions (90 and 270 m). The original forest cover data is 30 m resolution and takes values of 1 (forest) or 0 (no forest). We aggregated this data to 90 and 270 m resolution so that each grid cell in each year contains forest cover as a percentage (0–100%). In order to be able to detect change over time, we included forested grid cells with at least some (greater than 0) forest cover in the baseline year (2005). To control for the lack of independence of forested grid cells within the same CFM or MNP site, we also clustered standard errors at the site level. This step also addresses spatial autocorrelation between observations within the same site[51]. We also tested multilevel clustering of standard errors at the site and region level (22 administrative regions). Multilevel clustering did not affect our point estimates but rendered the observed differences in the years 2014–2017 marginally significant ($p < 0.1$ instead of $p < 0.05$).

**Tests of heterogeneity of impacts and spatial resolution**

Because the implementation of CFM is variable, we repeated the analysis for the sub-set of CFM for which the contracts were renewed, as this is an indicator that the CFM contracts were recognized and accepted by local communities and that implementation of CFM rules on the ground is more likely. Because geographic and socioeconomic context can influence both the effect of a political crisis and CFM performance, we explored how CFM performance varies based on the level of remoteness (measured as the distance from urban centers, roads, and villages), and population density. To explore this, we repeated the event study analysis, including all the same covariates described above but adding additional interaction terms representing each of these variables (Table 4).

We were also curious whether CFM performance might differ in areas with higher levels of development, as communities in such areas might be less dependent on forest resources. Similarly, we wondered if areas with higher levels of monitoring and enforcement might perform better. Therefore, we also ran the event study analysis with interaction terms for an indicator related to the level of development (an index of self-reported well-being indicators related to material assets) and an indicator related to the level of security (a self-reported indicator related to security conditions and risk of theft of property), both

measured at the level of *fokontany* (the smallest administrative unit within Madagascar) (Fig. S8). To explore the effect of spatial resolution on our results[66] we repeated all analyses at two spatial resolutions: 90 and 270 m.

## Reporting summary

Further information on research design is available in the Nature Portfolio Reporting Summary linked to this article.

## Data availability

The forest cover, deforestation, time-invariant, and time-variant covariates data generated in this study have been deposited in the Zenodo database under accession code https://doi.org/10.5281/zenodo.8132923. Shapefile polygons for protected areas in Madagascar are available from the World Database of Protected Areas: https://www.protectedplanet.net/country/MDG. The Community Forest Management areas polygon data are available under restricted access as this was what was agreed with the communities when the data was collected, access can be obtained upon reasonable request to Ranaivo Rasolofoson [ranaivo (dot) rasolofoson (at) duke (dot) edu].

## Code availability

Code used for this analysis can be found at: https://github.com/raenb0/madagascar. Code has been published to Zenodo (DOI: 10.5281/zenodo.10825817)[79].

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

## Acknowledgements
NSF Graduate Research Fellowship 2019067768 (RAN), Cornell Lab of Ornithology (RAN, ADR), Cornell University Department of Natural Resources and the Environment (RAN). Thanks to Ying Sun and Stephen Parry for help with the analysis. Thanks to: Catherine Kling, Jeffrey Milder, Tarana Chauhan, Seungmin Lee, Sergio Puerto, Kelsey Lynn Schreiber, Molly Jane Doruska, Elizabeth Tennant, Natasha Jha, Farnaz Safari Foroushani, Tess Lallemant, Harry Hyuk Son, Peizan Sheng, Dianna Tran, Trinh Pham, Leonel Borja Plaza, Kira Lancker, Dana Jezierny Smith; Julia P.G. Jones, Katie Devenish, Johanna Eklund, Richard Stedman, Bruce Lauber, Gloria Blaise, Charlie Tebbutt, Jessie Hughes, Jeanne Coffin-Schmitt, Viviana Ruiz Gutierrez, Courtney Davis, Andrew Stillman, Guillermo Duran, Teevrat Garg, Alexander Pfaff, Katie Devenish, Chris Golden, Robert Heilmayr, and Jorge Llopis for providing feedback on earlier drafts of this work.

## Author contributions
R.A.N., R.A.R., C.B.B. conceptualized and designed the analysis. R.A.N., R.A.R., C.B.B., G.V. contributed to the acquisition, analysis, and interpretation of the data. R.A.N. developed the code used in the analysis. R.A.N., R.A.R., C.B.B. drafted and R.A.N., R.A.R., C.B.B., G.V., A.D.R. substantially revised the manuscript. R.A.N. created the visuals. ADR supervised the study.

## Competing interests
The authors declare no competing interests.
