## [Peer Review File · Nature Communications]

The effect of a political crisis on performance of community forests and protected areas in MadagascarReviewer #1 (Remarks to the Author):

Neugarten et al studied the differences in deforestation between community and state- managed forests before and after political crisis and concluded that the changes in political crisis drove the different changes. There are two aspects of analysis determining the key message delivered in this manuscript: match balance and event study as subtitle, both terms seem meaningless, and I do not have much confidence in them. Moreover, what the practical actions drive such differences in deforestation after political crisis has not been studied and the conclusion is quite descriptive. Specifically,

1) the match balance is a statistical method that seems to control some covarying variables, but how this method control changes in disturbance or management over the temporal scale. This study is to detect changes in deforestation across time, this simple way is to identify changes in deforestation at the site/park level over time, rather than showing the only country-level average. By doing so, we can clearly spot where the distinct changes in deforestation happen, or whether there are significant changes in deforestation among the neighboring community and state- managed forests over political crisis periods.

2) Event study, I try to understand how to derive the table 2 using equation (2), I saw that there were parameters for each year but only one parameter for individual environmental variables. I first felt that they were calculated (regression) for each year, but where are the parameters for the other variables, from the supplementary, it convinces that they were regressed for each year. But when detecting the attribution over time, I would suggest to identify the relationships between them across time as a function of local park or sites. For instance, how the drought changes over time at a given park associated with how changes in deforestation. Using all the sampling across the spatial scale cannot support the attribution analysis over time and fail to capture temporal changes in deforestation associated with temporal changes drivers.

3) I suggest to control/isolate the impacts of temporal changes in these environmental variables provided in Table 4 on deforestation. Next, detecting changes in deforestation between both types of forest managements areas, if so, further determining whether they are related to political crisis.

Specific issues

L230-233 the decreases in woody cover seems very small (try to see if they are significant different) and can be caused by many aspects of reasons (extremes events). And should elaborate on how to convert such decreasing forest cover (percentage) into forest (area) loss?

Fig. 1 how did you get forest cover during 2005 or 2020, since Hansen et al only provided global forest cover map in 2000. Or what you show here only indicate deforestation?

Fig 4 what did the unit (%) stand for? Forest cover?

Fig. 2 and 4 the quality can be improved.

Reviewer #2 (Remarks to the Author):

This manuscript concerns a topic of high importance given that many of the world's most biodiverse and carbon-heavy forests grow in regions where there's political instability. This manuscript is noteworthy in offering a systematic analysis of the complex relationship between political instability and forest clearing. The writing is very clear, the models well described. Our greatest concern is with the limited attention to causal inference. Thus we recommend the manuscript be published on condition that the authors make a handful of essential but manageable revisions to bolster insights on

causality – or to at least be more transparent about uncertainty regarding causality (list A, below). We also recommend minor, optional improvements (list B).

List A: Necessary revisions

In the discussion (lines 288 and 313), the authors express uncertainty about whether forest policy in Madagascar changed during the study. This uncertainty raises doubt about possible confounding factors shaping deforestation. Given that some of the authors' have deep expertise in conservation policy in Madagascar, this 'maybe forest policy changed?' aspect is surprising. It ought to be possible to ascertain and rule out the possibility that there were substantive forest policy changes. E.g., by speaking to Malagasy conservation leaders/experts. Or taking time to dig deeper into the field-driven studies of deforestation in Madagascar during the period to at least offer examples.

The authors ought to acknowledge the distinction between tenure form and other factors influencing deforestation in community forest management (CFM), such as tenure security (see Robinson et al.'s "Does secure land tenure save forests? A meta-analysis of the relationship between land tenure and tropical deforestation" (Global Environmental Change 29, 2014). While we recognize data on tenure security is likely difficult if not impossible to obtain for all the CFMs in Madagascar, the authors should note this is a limitation of their study and that further research is needed to understand why crisis results in temporarily higher deforestation in CFM. We highlight tenure security in particular because the manuscript indicates that some CFMs have insecure tenure due to boundary conflicts with protected areas (lines 426-428), making this a particularly relevant aspect to address.

The authors ought to do more to help readers understand why deforestation was higher in CFMs compared to protected areas from 2014 to 2017, versus the absence of a significant difference in deforestation between CFM vs PA from 2018 to 2020. Again, how do local experts (Malagasy govt officials and NGO leaders) interpret these results? Was there no opportunity to share the results? What do other field-based analyses (albeit qualitative) offer?

Previous research at the national level on deforestation in protected areas during political crises (e.g., in Rwanda, DRC) reveals strong variation by region and individual protected area. (some parks actually experience reduced deforestation) Ideally the authors would add an appendix with descriptive information about deforestation rates in the various PAs, ideally also in CFMs. This would help make the paper more 'actionable' for practitioners/donors, etc. working in Madagascar.

There are some conspicuous gaps in the literature, e.g. Arakwiye, B et al. "Thirty years of forest-cover change in Western Rwanda during periods of wars and environmental policy shifts." *Regional Environmental Change* 21 (2021): 1-15. and Ordway, Elsa M. "Political shifts and changing forests: Effects of armed conflict on forest conservation in Rwanda." *Global Ecology and Conservation* 3 (2015): 448-460.

Add a couple sentences to explain the meaning of 'concerted efforts'? (line 283). Relatedly: the authors acknowledge that logging is difficult to measure via remote sensing, yet they emphasize that park managers could not control logging. Could they not control agriculture expansion either?

Part B: Minor revisions:

While the authors' models are robust, the manuscript could be improved by clarifying why certain assumptions were made in the methods section (detailed below).

- Line 264-265 and lines 383-388: Renewed contract could be a sufficient proxy for CFM implementation, but more background information on what a renewed contract entails is needed (duration of contracts etc., process to renew) to assesses the validity of using this sub-set to test

heterogeneity.

- Ideally include a time lag variable for CFMs (how long since contract secured/how long have CFM been operating) to better understand if this impacts the ability of CFM to prevent deforestation.
- Clarify why 2005 is used as the baseline year. Does conducting the matching in 2005 instead of right before the crisis creates any complications in the validity of the matching technique.
- Because of the matching technique, the authors remove points where there was overlap between the PA and CFM boundaries. Because this indicates tenure insecurity for the WHOLE CFM we ask the authors to consider whether they should remove all points within these CFM from the analysis since the literature suggests those CFMs would not be able to control the boundaries of their forest even in times of stability. Even if the authors do not wish to remove them from the main model, we suggest they create a sub-set with these CFMs removed. Authors also mention that CFMs that do not allow for commercial use result in lower deforestation (line 121-122), and we wonder whether the authors are able to incorporate differences in CFM contracts in their analysis.
- The authors explain why they choose event study design instead of DID, but we suggest the authors consider including the DID model to compare results, as is custom within the conservation impact literature.
- Did the presence of an NGO have any effect?
- Do the authors have any insights from their research on what could be done to prevent a similar spike in deforestation after the crisis? I.e., given the urgency of the issue, offering more explicit recommendations for practitioners and future researchers would be appropriate.

Reviewer #3 (Remarks to the Author):

[Editor's note: Reviewer 3 co-reviewed the manuscript with one of the reviewers who provided the listed reports as part of the Nature Communications initiative to facilitate training in peer review and appropriate recognition for co-reviewers]

REVIEWER COMMENTS

Reviewer #1 (Remarks to the Author):

Reviewer: Neugarten et al studied the differences in deforestation between community and state-managed forests before and after political crisis and concluded that the changes in political crisis drove the different changes. There are two aspects of analysis determining the key message delivered in this manuscript: match balance and event study as subtitle, both terms seem meaningless, and I do not have much confidence in them.

- Response: We have tried to do a better job explaining terms and methods which may be unfamiliar to some reviewers and readers. Both statistical matching and event study methods are well established and widely used to evaluate the impact of interventions across fields including policy¹, education^{2,3}, and public health^{4,5} interventions. There are a large number of peer-reviewed publications⁶⁻⁸ and textbooks^{9,10} explaining how the methods work and how they can be appropriately applied. There are also a large number of publications which have used these methods to evaluate the effects of biodiversity conservation interventions, specifically¹¹⁻³⁰.
- The basic concept underlying both methods is that they help isolate the effect of an event (in our case, the political crisis) from other factors. Statistical matching allows us to identify forested areas that are similar, on average, within community forest management (CFM) and state protected areas administered by Madagascar National Parks (MNP). You can think of these as “treatment” (CFM) and “control” (MNP) sites. The problem is that in non-experimental, observational data like those used in this study, those sites will typically differ in many dimensions, not just in the dimension of interest, in our case the conservation status of each sample point (90 m forest grid cell). So one matches on a range of characteristics so as to generate more directly comparable pairs, which is referred to as “match balance”. Figure 3 in the manuscript illustrates how different CFM and MNP pixels are from one another on average, before matching (red dots), and how the matching achieves this balance (blue triangles).
- Event study allows us to compare outcomes before, during, and after the political crisis, and control for other temporal events (climate events, rice price fluctuations, etc.) that might influence deforestation outcomes over time. When combined, the two methods allow us to attribute any remaining differences in performance to the political crisis. You can think of this as a form of a before/after/control/impact (BACI) study design, which may be more familiar. The statistical matching allows us to compare “control” and “treatment” sites, and the event study allows us to compare trends before and after the crisis. In combination, the two methods provide greater confidence that the observed differences are due to the crisis, and not some other confounding factors.
- We have re-organized and revised the relevant sections in the Methods section to explain and justify our use of both methods more clearly and have provided additional citations. Modified lines 169-188:

“To allow causal interpretation of our results, we used deforestation data derived from remote sensing^{31,32} and a counterfactual approach implemented through a combination of statistical matching and an event study design. We combined two methodological approaches to control for factors that can confound the estimated relative performance of CFM and MNP. First, we used statistical matching³³ to identify forest areas within CFM and MNP that are similar across a range of observed biophysical and geographic confounding characteristics, such as remoteness and suitability for agriculture. This allows us to make an “apples-to-apples” comparison of similar forested sites within CFM and MNP, controlling for time-invariant confounding factors. Second, we conducted an event study analysis⁸ to control for all relevant observed time-variant confounding factors, such as rice prices and climate variables. The event study also allows us to control for any differences in deforestation trends in CFM and MNP in the pre-crisis period, to isolate the effect of the crisis. Additionally, an event study design allowed us to examine the yearly variation in the effect of the crisis on the relative performance of CFM compared to MNP. We also explored the effects of spatial resolution on our results. Lastly, because the impacts of the crisis on CFM performance may vary as a function of contextual variables, we explored the moderating effects of contextual factors such as distance to cities, distance to roads, and population density. In our study, the “event” is the onset of the crisis, and our specific research question is “What was the effect of the crisis and post-crisis period on relative performance of CFM and MNP, in terms of their ability to reduce deforestation?”

- Methods section, “Statistical matching” (lines 432-455) now reads:

“The goal of matching is to identify and control for observed confounding factors^{6,30,33}, in our case, variables that influenced the likelihood that a site was designated as CFM or MNP and also influence the probability of deforestation. The matching step allowed us to identify a matched sample which included a “treatment” group (CFM forest grid cells) and a “comparison” group (matched MNP forest grid cells that were statistically similar to the CFM grid cells) that we could use for an “apples-to-apples” comparison. As variables for matching, we therefore included data on factors that are known to influence both the probability that a site is designated as a CFM or MNP area, and also the likelihood of deforestation (Table 3)²². These include suitability for rice agriculture³⁴, elevation and slope³⁵, average annual precipitation 1970-2000³⁶, distance to forest edge in 2005 (calculated for this analysis), distance from roads, cart tracks, villages, and urban centers²², population density in 2005³⁷, and vegetation zones²². Because the goal of matching is to identify variables which might have influenced site designation as CFM or MNP, all variables used in matching are for the first year in the study period (2005) or earlier (such as historic precipitation averages) or were time-invariant (such as elevation and slope).

We conducted 1:1 genetic matching, with replacement. Genetic matching is not a unique matching method, rather it identifies a method (such as propensity score matching or matching based on Mahalanobis distance) which optimizes covariate balance³⁸. We used

the “MatchIt” package in R³⁹ and covariate balance was assessed using the cobalt package⁴⁰. We also tested two alternative matching methods, propensity score matching and Mahalanobis distance matching (Fig. S6) but genetic matching led to better balance in the covariates.”

- The Methods section, “Event study” (lines 493-528), now reads:

“After matching, we conducted an event study analysis to investigate the effect of the political crisis (the event), i.e. compare CFM and MNP performance before, during, and after the political crisis. The goal of the event study analysis is to control for all time-invariant and observed time-variant confounding factors which may influence deforestation outcomes⁴¹ differently in CFM and MNP. We also performed a two-period difference-in-differences (DiD) analysis, comparing the pre-crisis period (2005-2009) and the crisis period (2010-2014) (Supplemental Materials). We note that our event study model is a more general form of a DiD model that provides two advantages relative to DiD. It allows us to control for any differences in deforestation trends in CFM and MNP in the pre-crisis period. And it allows us to explore yearly variation of relative performance of CFM and protected areas (instead of only two time periods, pre- and post-crisis, in the case of a DiD). Thus, we focus on the event study here; results of the DiD are included in the Supplemental Materials. In both the DiD and event study analysis, we controlled for annual rice prices and rice price volatility⁴², annual climate variables such as maximum precipitation, drought severity, and maximum wind speed (an indicator of cyclones)⁴³, annual population density²², and annual distance to forest edge (Table 4, Fig. S7). All variables were spatially explicit (Fig S7) and were included on an annual basis for the years 2005-2020. We also controlled for each “year” so as to capture any other time-variant factors which could influence deforestation outcomes between years but that would be common to all pixels in the study. In addition to their potential for influencing deforestation outcomes, the selection of covariates was also influenced by data availability, as our study design requires data that are both temporally and spatially comprehensive (that is, available annually from 2005-2020, and for both CFM and MNP areas).

If deforestation in CFM was significantly different than deforestation in matched MNP areas during the political crisis, after controlling for time-variant factors (such as rice prices or drought severity), we can attribute the difference to the interacting effect of the treatment (CFM vs. MNP) and the political crisis. For the event study analysis, we used an OLS regression with interaction terms representing the year (2005-2020), years post crisis, and fixed effects for each spatial unit, using the “fixest” package in R⁴⁴. Our event study model takes the form:

$$Y_{it} = \beta_1 CFM_i + \tau_1 year_t + \tau_2 year_t CFM_i + \gamma YearsPostCrisis_t + \delta CFM_i YearsPostCrisis_t + \psi X_{it} + \mu_i + \varepsilon_{it} \quad (2)$$

Where Y_{it} is forest cover loss (percentage) in each forest grid cell i in year t ($t = 2005-2020$); $CFM = 1$ if the grid cell falls within a CFM and $CFM = 0$ if it falls within an MNP; $YearsPostCrisis = 0$ for the crisis years (2005-2009) and then 2010, 2011, and so on; X_{it} is a vector of time-variant controls (Table 4); and ε_{it} is a random error term. We included individual fixed effects for each forest grid cell (μ_i), and clustered standard errors at the site level (where each site is a unique CFM or MNP).”

Reviewer: Moreover, what the practical actions drive such differences in deforestation after political crisis has not been studied and the conclusion is quite descriptive. Specifically,

1) the match balance is a statistical method that seems to control some covarying variables, but how this method control changes in disturbance or management over the temporal scale. This study is to detect changes in deforestation across time, this simple way is to identify changes in deforestation at the site/park level over time, rather than showing the only country-level average. By doing so, we can clearly spot where the distinct changes in deforestation happen, or whether there are significant changes in deforestation among the neighboring community and state-managed forests over political crisis periods.

- Response: Regarding “how this method controls changes in disturbance or management over the temporal scale”, As noted by the reviewer, we’ve used statistical matching to control for a wide range of confounding variables that may affect deforestation, though we see that we failed to sufficiently explain how that approach also addressed temporal changes. We specifically controlled for changes in disturbance over time in the event study model, which controls for time-varying confounding factors (climate-related variables, rice prices, changes in human population density, changes in distance from forest edge, etc.) We also include a vector of indicator (i.e., binary) variables for “year” which should control for any other factors that vary between years but are common to all forest sample points, both those in CFM and in MNP.
- We’ve added the following text to the Results section to clarify that point:
- “Importantly, a simple comparison of deforestation rates does not control for confounding variables that both affect forest loss and influence the likelihood that a site is designated as a CFM or MNP. In Madagascar, confounding variables include distance from the nearest road, distance from the nearest village, distance from the nearest urban center, distance from forest edge, slope, elevation, and agricultural suitability^{34,39}. Also, dynamic events such as climate extremes (droughts, floods, cyclones) and price fluctuations (such as global rice prices and price volatility)⁴⁸ could differentially affect deforestation in CFM and MNP areas. Therefore, we conducted statistical matching followed by an event study design to control both time-invariant and time-variant confounding variables.”
- Regarding the reviewer’s comment that we should “identify changes in deforestation at the site/park level over time, rather than showing the only country-level average”. As explained above, a simple comparison of deforestation in CFM and MNP sites would not control for factors such as differences in location which affect performance. Some CFM sites may have very high rates of deforestation because they are located close to roads

and urban areas, and therefore they may be avoiding even higher rates of deforestation that would be expected of forest sample points with such characteristics. In contrast, some sites may have low deforestation but because they are located in very remote areas they would have very low deforestation even without protection. In this analysis we are trying to separate raw deforestation outcomes from performance (where performance is defined as “avoided” deforestation – that is, how much deforestation was avoided by the conservation intervention, relative to a precise, quantifiable counterfactual.)

- Our study focuses on how the crisis affected the performance of CFM and MNP at the country level. That said, we agree that estimating performance at the site level would add an interesting dimension to this study, but unfortunately the small number of MNP (45) relative to CFM (362) along with systematic differences between them (CFM tend to be closer to roads, urban areas, etc.) prevents us from identifying good matches at the site level. This is why we used forest grid cells as the unit of analysis, as it is possible to identify forest grid cells in CFM and MNP that have similar characteristics (good matches).
- We did, however, explore heterogeneity of impacts across sites using site attributes such as distance to roads, cities, etc. As explained in the Results section:
- “We found that CFM further from urban centers performed better than those closer to cities in the post-crisis period, and the difference was statistically significant in 2015, 2016, and 2018 (Fig. S10, Table S5). Thus, distance from urban centers, a measure of remoteness, appears to influence the performance of CFM. Even in remote areas, however, CFM were less effective at reducing deforestation than similarly remote MNP. We explored potential heterogeneity of effects using other variables, including distance from roads, distance from villages, population density, level of development, and security, but found no consistent or significant effect of any of these variables (Tables S6-S8).”

Reviewer: 2) Event study, I try to understand how to derive the table 2 using equation (2), I saw that there were parameters for each year but only one parameter for individual environmental variables. I first felt that they were calculated (regression) for each year, but where are the parameters for the other variables, from the supplementary, it convinces that they were regressed for each year. But when detecting the attribution over time, I would suggest to identify the relationships between them across time as a function of local park or sites. For instance, how the drought changes over time at a given park associated with how changes in deforestation. Using all the sampling across the spatial scale cannot support the attribution analysis over time and fail to capture temporal changes in deforestation associated with temporal changes drivers.

- Response: We appreciate your suggestions and have tried to clarify the relationship between table 2 and Equation 2. You are correct that a single matrix, X_{it} refers to each forest sample point’s observations of a vector of time-variant control variables, which are described in Table 4. Those have a corresponding parameter vector, ψ , that we estimate directly along with the other parameters specified in Equation 2. Summarizing multiple

covariates using a single variable in the model equation is a standard notation in statistics and econometrics.

$$Y_{it} = \beta_1 CFM_i + \tau_1 year_t + \tau_2 year_t CFM_i + \gamma YearsPostCrisis_t + \delta CFM_i YearsPostCrisis_t + \psi X_{it} + \mu_i + \varepsilon_{it} \quad (2)$$

As we explain:

“Where Y_{it} is forest cover loss (percentage) in each forest grid cell i in year t ($t = 2005-2020$); $CFM = 1$ if the grid cell falls within a CFM and $CFM = 0$ if it falls within an MNP; $YearsPostCrisis = 0$ for the crisis years (2005-2009) and then 2010, 2011, and so on; X_{it} is a vector of time-variant controls (Table 4); and ε_{it} is a random error term. We included individual fixed effects for each forest grid cell (μ_i), and clustered standard errors at the site level (where each site is a unique CFM or MNP).”

- The reviewer is correct that the value of each time-variant covariate was calculated for every observation (forest grid cell) (i) and in each year (t) in the event study. The notation provides a shorthand way to describe this without listing every single parameter.
- Results include a separate coefficient for each year because the vector “year” includes one binary indicator variable for each year in Equation 2. Continuous variables, on the other hand, are associated with only a single coefficient.
- In the Supplemental Materials, we provide examples of the spatial datasets (maps) used in the analysis (Figure S7). Though we provide only one example of each dataset from a single year (2005), the full analysis includes data from 2005-2020. The complete datasets used in the analysis are also provided for download via Zenodo: <https://zenodo.org/record/8132923> - as you can see from the site, there are many rasters there, corresponding to all variables and all years 2005-2020.

Reviewer: 3) I suggest to control/isolate the impacts of temporal changes in these environmental variables provided in Table 4 on deforestation. Next, detecting changes in deforestation between both types of forest managements areas, if so, further determining whether they are related to political crisis.

- Response: We did control for the impacts of temporal changes in all variables listed in Table 4, just as the reviewer suggests. We have revised the Methods, “Event study” section to explain this more clearly (see above.)

Reviewer: Specific issues

L230-233 the decreases in woody cover seems very small (try to see if they are significant different) and can be caused by many aspects of reasons (extremes events). And should elaborate on how to convert such decreasing forest cover (percentage) into forest (area) loss?

- Response: We recognize that the estimated effect might seem small, but it becomes ecologically meaningful when one considers the amount of forest under management by

CFM before the crisis (475,333 ha, Table S1). We have clarified this in the Results, Event study section:

- The difference in the effect of CFM relative to MNP on deforestation in the years 2014-2017 ranged from $1.7 \pm 1.4\%$ per year to $2.4 \pm 1.0\%$ per year. In other words, CFM had higher annual deforestation than MNP in those years, even after controlling for differences in location and other confounding variables. In the year immediately preceding the crisis (2008), CFM contained 475,333 ha of forest (Table S1). Thus CFM forests lost an estimated $8,103 \pm 6,435$ ha/year to $11,532 \pm 9,508$ ha/year more tree cover than similar forests in MNP. This is equivalent to a total of $36,483 \pm 28,775$ ha for the 2014-2017 period (approximately 51,000 \pm 40,000 soccer fields).
- As an aside, similar effect sizes (<5%) have been estimated from quasi-experimental studies (Ribas et al. 2020). We have also added an explanation and citation in the Discussion:
 - “The estimated difference in performance between CFM and MNP in the post-crisis years may seem small (1.7-2.4%/year) but given that CFM contained nearly half a million hectares of forest at the start of the crisis, such a difference is ecologically meaningful. Also, our estimated effect sizes are in line with other studies that use quasi-experimental methods to estimate effects of protected areas on avoided deforestation, which are typically under 5%, and often under 1%¹⁸.”

Reviewer: Fig. 1 how did you get forest cover during 2005 or 2020, since Hansen et al only provided global forest cover map in 2000. Or what you show here only indicate deforestation?

- Response: We used 2000 forest cover from a previously published study (Vieilledent et al. 2018). Then, we used Hansen et al. 2013 tree cover change estimates to calculate change in forest cover for each year from 2001-2021. So for example, a pixel that was considered forested in 2000 according to Vieilledent et al, but then was considered to have experienced tree cover loss in 2001 according to Hansen, would be considered to be deforested. The reason we combined the products is that Hansen forest cover in 2000 is reported as a percentage (e.g. 0-100% forest cover) and the user must select a threshold (e.g. 50% forest cover) to define “forest” versus “non-forest”. Therefore we relied on Vieilledent et al. 2018 to define forest and non-forest areas in 2000, as this product is already validated and published. We made all the data used in the analysis available via Zenodo: <https://zenodo.org/record/8132923>
- In the Methods, “Deforestation” section, we explain:
 - “To calculate deforestation, we used forest cover data for Madagascar for the year 2000³² combined with global tree cover change estimates³¹ to generate an annual time series (Fig. S2). The 2000 forest cover product was produced based on satellite imagery (Landsat TM and ETM+)⁴⁵ combined with a 2000 tree cover percentage map³¹ to fill gaps due to the presence of clouds³².”

Reviewer: Fig 4 what did the unit (%) stand for? Forest cover?

- Response: Yes, the Y-axis represents the effect of the political crisis on annual deforestation (percent tree cover loss per year). We have clarified this in the Figure 4 caption.

Reviewer: Fig. 2 and 4 the quality can be improved.

- Response: We have provided high-resolution versions of all Figures as separate attachments with our revised submission.

Reviewer #2 (Remarks to the Author):

This manuscript concerns a topic of high importance given that many of the world's most biodiverse and carbon-heavy forests grow in regions where there's political instability. This manuscript is noteworthy in offering a systematic analysis of the complex relationship between political instability and forest clearing. The writing is very clear, the models well described. Our greatest concern is with the limited attention to causal inference. Thus we recommend the manuscript be published on condition that the authors make a handful of essential but manageable revisions to bolster insights on causality – or to at least be more transparent about uncertainty regarding causality (list A, below). We also recommend minor, optional improvements (list B).

- Response: Thank you for your comments regarding the importance of the topic, clear writing, and model description. We thank the reviewer for the detailed suggestions regarding improving clarity about causal inference and associated uncertainty, as well as the recommended minor improvements. We have addressed all comments in this letter as well as our substantially revised and strengthened manuscript.

List A: Necessary revisions

In the discussion (lines 288 and 313), the authors express uncertainty about whether forest policy in Madagascar changed during the study. This uncertainty raises doubt about possible confounding factors shaping deforestation. Given that some of the authors' have deep expertise in conservation policy in Madagascar, this 'maybe forest policy changed?' aspect is surprising. It ought to be possible to ascertain and rule out the possibility that there were substantive forest policy changes. E.g., by speaking to Malagasy conservation leaders/experts. Or taking time to dig deeper into the field-driven studies of deforestation in Madagascar during the period to at least offer examples.

- Response: Thank you for this comment and suggestions. While we are, to our knowledge, the first to document a spike in overall deforestation across the country following the 2009-2014 political crisis, and we acknowledge that we cannot conclusively identify the

cause. Based on your suggestion, we reviewed relevant documents and corresponded with several conservation leaders and experts from Madagascar to supplement our team's (significant) personal knowledge of conservation policy in the country over this period. We found no evidence of changes in forestry policy following the crisis (2014-2017) and, consequently, have removed mention of policy change as a possible explanation from the discussion.

- There were several alternative theories raised by experts we spoke to about what might have caused the observed spike in deforestation after the crisis, and the difference in performance between CFM and MNP during the post-crisis era, which we hope to explore in future work. These theories are described in the discussion and include:
 - a) A decline in international funding for conservation which began during the crisis^{46,47}, but the effects likely lingered and/or became more severe for several years after the crisis officially ended;
 - b) The effect of political elections⁴³ in late 2013 (which ended the crisis) and 2018;
 - c) Weakened governance and increased corruption during the crisis, with effects that lingered and/or became more severe in the post-crisis period.
- We have revised the first paragraph of the discussion to include these possible explanations:

“We found that, despite conservation efforts which sought to protect forests during Madagascar’s recent political crisis, annual rates of deforestation accelerated at the end of the crisis – a phenomenon that, to our knowledge, has not been reported previously. Understanding the cause of this post-crisis increase in deforestation is beyond the scope of this analysis, but we can provide some theories that could be explored in future work. One possibility is that we detected a lagged response to events that occurred during the crisis. Funding for conservation declined precipitously during the crisis^{2,51}, and it took several years for financial support to be restored to pre-crisis levels. Weak governance and increased corruption during the crisis^{2,46} might have had lingering effects or become more severe in the post-crisis period. Some other theories we explored are that the return to political stability in the post-crisis period might have initiated a change in forestry policy, or triggered an increase in economic activity, putting even more pressure on forests. We found no evidence of a formal change in forestry policy during the post-crisis era, however, and per-capita GDP did not increase substantially during 2014-2017³⁰. What drove the observed post-crisis deforestation spike therefore requires further study.”

Reviewer: The authors ought to acknowledge the distinction between tenure form and other factors influencing deforestation in community forest management (CFM), such as tenure security (see Robinson et al.'s "Does secure land tenure save forests? A meta-analysis of the relationship between land tenure and tropical deforestation" (Global Environmental Change 29, 2014). While we recognize data on tenure security is likely difficult if not impossible to obtain for all the CFMs in Madagascar, the authors should note this is a limitation of their study and that further research is needed to understand why crisis results in temporarily

higher deforestation in CFM. We highlight tenure security in particular because the manuscript indicates that some CFMs have insecure tenure due to boundary conflicts with protected areas (lines 426-428), making this a particularly relevant aspect to address.

- Response: We thank the reviewer for this suggestion. We have edited the text (Methods, “Limitations” section) to describe how lack of existing data on tenure security is indeed a limitation. In the Discussion section, we also added references to Robinson et al. 2014⁴⁸, Rakotonarivo et al. 2023⁴⁹ on how the lack of land tenure security in Madagascar complicates forest restoration projects, and to Montagne et al. 2013⁵⁰ who explain that the law establishing CFM includes a provision through which tenure security can be transferred, but there have been few cases in which this has actually occurred. (And just a small note that we had already cited Hajjar et al. 2021⁵¹ which found that community forests with secure land tenure rights were more likely to have successful ecological and socioeconomic outcomes.)
- If lack of tenure security explained the poor performance of CFM overall, we would have expected to find (but did not) a consistent signal before, during, and after the political crisis. To our knowledge, there was no change in tenure security after the crisis ended. If such a change had occurred, it could have helped explain the observed difference in CFM performance during that period.
- We agree that boundary conflicts between CFM and protected areas are an indicator of tenure insecurity; as described in the Methods section, we excluded such areas from our analysis.

Reviewer: The authors ought to do more to help readers understand why deforestation was higher in CFMs compared to protected areas from 2014 to 2017, versus the absence of a significant difference in deforestation between CFM vs PA from 2018 to 2020. Again, how do local experts (Malagasy govt officials and NGO leaders) interpret these results? Was there no opportunity to share the results? What do other field-based analyses (albeit qualitative) offer?

- Response: We agree with the reviewer that the difference in CFM and PA performance 2018-2020, though not statistically significant, is intriguing (see Fig 4 and Table 2.) and suggests that that CFM continued to perform worse than MNP, even after controlling for both time-invariant and time-variant confounding factors. The absence of a statistically significant difference in deforestation between CFM vs PA from 2018 and 2020 might indicate that the situation was beginning to stabilize, though the time period is brief. The magnitude of the estimated difference between CFM and MNP remains elevated but becomes less precisely estimated the further one moves from the crisis. That would be consistent with differential rates of recovery among CFM, some of which began to converge back towards MNP within a few years of the crisis' end, others of which continued to lag far behind. Given heterogeneous management of CFMs and relatively more homogeneous management of MNP sites, that makes sense to us. We have now

mentioned this in the Results and Discussion sections, so we thank the reviewer for pointing this out.

- To our knowledge, our study is the first to document the temporary increase in deforestation in the post-crisis era, so previous studies haven't been helpful in terms of understanding more recent deforestation dynamics. For example, one recent study (Suzzi-Simmons 2023⁵²) summarizes total national deforestation 2000-2021, but doesn't break down deforestation data by year. As mentioned above, conversations with local experts led to a number of theories, but additional work is needed to validate possible explanations.

Reviewer: Previous research at the national level on deforestation in protected areas during political crises (e.g., in Rwanda, DRC) reveals strong variation by region and individual protected area. (some parks actually experience reduced deforestation) Ideally the authors would add an appendix with descriptive information about deforestation rates in the various PAs, ideally also in CFMs. This would help make the paper more 'actionable' for practitioners/donors, etc. working in Madagascar.

- Response: We provide total deforestation data in CFM, MNP, and nationally in Table 1 and annual data in the Supplemental Material (Table S1), but do not summarize the data by site. Individual sites are subject to a variety of factors (e.g. slope, elevation, distance to roads, cities, population density, suitability for agriculture, etc.) that would complicate interpretation. In essence, reporting CFM-specific and MNP specific deforestation rates invites precisely the sort of confounded comparisons we work hard to avoid with the statistical matching and event study model. Controlling for differences in location as well as other site attributes is one of the motivations underlying the current analysis. Further, there are 45 MNP and 362 unique CFM sites in our sample, each with 15 years of data (2005-2020), requiring a table with 6,105 cells of estimates, which seems very difficult for readers to digest.

Reviewer: There are some conspicuous gaps in the literature, e.g. Arakwiye, B et al. "Thirty years of forest-cover change in Western Rwanda during periods of wars and environmental policy shifts." *Regional Environmental Change* 21 (2021): 1-15. and Ordway, Elsa M. "Political shifts and changing forests: Effects of armed conflict on forest conservation in Rwanda." *Global Ecology and Conservation* 3 (2015): 448-460.

- Response: We have added both citations to the introduction, thank you. Our focus in this analysis was not on the impacts of a crisis (such as armed conflict) on deforestation in general, but rather which kinds of conservation interventions are resilient (or vulnerable) during a crisis. Studies of the impacts of armed conflict on deforestation typically do not control for other confounding factors which influence conservation performance, such as location differences or climate-related variables. Thus we had originally focused on the few examples of such studies we could find, which include studies from Nepal⁵³, Colombia⁵⁴, and Sierra Leone⁵⁵. Nonetheless, we have included both of these studies in the Introduction:

- “There have been very few counterfactual-based studies which investigate the effectiveness of conservation interventions in times of crisis. The few examples we identified focused on armed conflict. In Nepal, local institutions were able to organize and cooperate to reduce forest fragmentation even during periods of violent conflict⁵³. In Colombia, large protected areas were more effective at reducing deforestation during periods of conflict between the government and guerilla fighters⁵⁴. In Sierra Leone, armed conflict was linked to lower rates of deforestation, but the performance of conservation interventions was not specifically analyzed⁵⁵. Two studies from Rwanda found that armed conflict led to increased deforestation^{56,57}, but did not control for potential confounding factors such as location or climate-related variables.”

Reviewer: Add a couple sentences to explain the meaning of ‘concerted efforts’? (line 283). Relatedly: the authors acknowledge that logging is difficult to measure via remote sensing, yet they emphasize that park managers could not control logging. Could they not control agriculture expansion either?

- Response: Apologies for being unclear, by “concerted efforts” we were referring broadly to conservation efforts (protected areas, CFM) which seek to avoid deforestation in Madagascar, we have revised this sentence to read:
- “We found that, despite conservation efforts which sought to protect forests during Madagascar’s recent political crisis, annual rates of deforestation accelerated at the end of the crisis – a phenomenon that, to our knowledge, has not been reported previously.”
- And yes, the reviewer is correct that the ability of park managers to control illegal logging and other forms of forest clearing (such as for agriculture) was low or nonexistent during the crisis. Our data (based on remote sensing) clearly indicates that there was forest cover loss taking place within MNP before, during, and after the political crisis. This has also been established by field-based studies such as Llopis et al. 2019⁵⁸. We have clarified this in the introduction and discussion.

Reviewer:

Part B: Minor revisions:

While the authors’ models are robust, the manuscript could be improved by clarifying why certain assumptions were made in the methods section (detailed below).

- Line 264-265 and lines 383-388: Renewed contract could be a sufficient proxy for CFM implementation, but more background information on what a renewed contract entails is needed (duration of contracts etc., process to renew) to assesses the validity of using this sub-set to test heterogeneity.
 - Response: Thank you for this comment. We have added the following explanation to the Methods:

- “The degree to which CFM contract terms are implemented in practice likely varies among sites. Unfortunately, consistent data on implementation is not available for the majority of CFM sites. As mentioned above, after an initial three-year period, if all parties agree that the CFM is being properly managed, the contract can be renewed for an additional ten-year period (Raik and Decker 2007). Therefore, we use contract renewal as an indicator that the CFM contracts were being implemented in practice and were not agreements “on paper” only. We repeated our sampling within a sub-set of CFM contracts that were renewed.”

Reviewer: Ideally include a time lag variable for CFMs (how long since contract secured/how long have CFM been operating) to better understand if this impacts the ability of CFM to prevent deforestation.

- Response: While we agree that ‘time since contract was secured’ is a relevant variable, for our analysis, we restricted our analysis to CFM that were established prior to 2005 to provide at least four years of baseline/pre-crisis. The first CFM contracts were signed in 1999, therefore all CFM included in our analysis were established between 1999 and 2005. This was because we wanted at least four years of deforestation data in comparable CFM and MNP sites before the crisis started (2005-2009) to establish a sufficient baseline of pre-crisis deforestation trends. This allowed us to evaluate (and control for differences) in trends in deforestation before the crisis started, which was necessary for our event study design. As a result, most of the CFM sites should have been established within a relatively short timeframe. More sites were established in the later part of this period (as CFM contracts became more widely established.) For reference, we summarized the number of CFM sites included in our sample and the years in which the contracts were signed:

Year signed	Count of sites
1999	7
2000	19
2001	25
2002	58
2003	88
2004	54
2005	111
Total	362

Reviewer: Clarify why 2005 is used as the baseline year. Does conducting the matching in 2005 instead of right before the crisis creates any complications in the validity of the matching technique.

- Response: Thank you for this question. As mentioned above, we used 2005 as the baseline year in order to establish several years of pre-crisis deforestation trends, which allowed us to control for differences in CFM and MNP deforestation in the pre-crisis period in our event study analysis. Since many CFM were established between 2000 and 2005, it didn't make sense to use an earlier baseline year (e.g. 2000, when annual high-resolution deforestation data is first available) as many CFM sites did not exist in 2000. We have clarified this in the Methods section, which already explained ("Sampling", first sentence):
 - "To establish forest trends during a baseline (pre-crisis, 2005-2009) period, we focus on 362 CFM sites established beginning with the first CFM contract in 1999 through 2005, and 45 protected areas managed by MNP also established prior to 2005."
 - We added: "We use 2005 as the baseline year as it provides a sufficient pre-crisis baseline, and because many CFM sites were established between 2000 and 2005."
- Regarding using 2005 as the matching year, the goal of matching is to control for confounding factors which influence both the likelihood that a site was designated as CFM versus MNP, and also influence deforestation outcomes. Therefore we used time-invariant attributes (for example, elevation, slope) and also attributes that related to site characteristics prior to the start of the analysis (average annual precipitation in the period prior to 2005, human population density in 2005, distance from roads in 2005, etc.) The logic is that such attributes might have influenced the likelihood that the site was designated as CFM or MNP, and therefore were in place prior to the site designation (2005 or earlier.) We explained this in the Methods section:
 - "The goal of matching is to identify variables that influenced the likelihood that a site was designated as CFM or MNP and also influence the probability of deforestation. Therefore, all variables used in matching are for the first year in the study period (2005) or earlier (such as historic precipitation averages) or were time-invariant (such as elevation and slope)."

Reviewer: Because of the matching technique, the authors remove points where there was overlap between the PA and CFM boundaries. Because this indicates tenure insecurity for the WHOLE CFM we ask the authors to consider whether they should remove all points within these CFM from the analysis since the literature suggests those CFMs would not be able to control the boundaries of their forest even in times of stability. Even if the authors do not wish to remove them from the main model, we suggest they create a sub-set with these CFMs removed.

- Response: This is a good suggestion, thank you. We did include a separate analysis on a sub-set of CFM for which contracts were renewed. These CFM do not overlap with MNP boundaries (see below map). Our results for this sub-set of CFM were the same as for all CFM, which gives us confidence that our results for the sub-set of CFM with no overlap of MNP would give us the same result.

-  CFM that were renewed
-  Madagascar National Parks (MNP)

Reviewer: Authors also mention that CFMs that do not allow for commercial use result in lower deforestation (line 121-122), and we wonder whether the authors are able to incorporate differences in CFM contracts in their analysis.

- Response: Unfortunately, data on commercial use are not available for the vast majority of CFM. The cited study (Rasolofoson et al. 2015²²) did indeed find that deforestation was lower within CFM that prohibit commercial use, but this was based on data from four sites around the country (see Fig 1 in that paper). The co-authors discussed repeating such an analysis in the current paper but agreed that the sample size of CFM for which we have data on commercial use is too small. We hope that better data on CFM contract terms and management becomes available in the future.

Reviewer: The authors explain why they choose event study design instead of DID, but we suggest the authors consider including the DID model to compare results, as is custom within the conservation impact literature.

- Response: thank you for this suggestion. We included the results of a difference-in-differences (DiD) analysis, using two periods: pre-crisis (2005-2009) and crisis (2010-2014), in the Supplemental Materials (S2 and Table S10). The results are consistent with the event study model results. We note that our event study model is a form of a DiD model, but has two advantages: it controls for differences in deforestation trends in CFM and MNP in the pre-crisis period, and it allows us to explore variation in effect of the crisis over time (annually, instead of only in two periods.) We clarified this in the main text (Methods, “Event Study” section).

Reviewer: Did the presence of an NGO have any effect?

- Response: we did not specifically include information on the presence of an NGO. We are not sure if the reviewer is asking about NGOs supporting CFM or NGOs managing protected areas (or both). Regarding whether NGOs supporting CFM might have had an effect, this is quite plausible. Unfortunately, we do not have consistent data on which CFM were supported by NGOs. Rasolofoson (2015) suggested that one possible reason for better performance of CFM that did not allow commercial harvest could be because those CFM receive support from NGOs, for example “direct payments for conservation.” This would be consistent with the theory that worse performance of CFM in the post-crisis period could be due to reduced levels of NGO funding. Anecdotally, during and after the crisis, levels of funding and support for CFM management declined. We also don’t currently have data on funding for CFM or MNP, although we hope such information will be available in the future.
- As for protected areas, as the reviewer may be aware, there are examples of protected areas (PAs) managed by NGOs (Makira, CAZ, COFAV) but we chose not to include them for two key reasons (1) These PAs have not been established until very recently. Makira, the most advanced of those PAs, was established as a national park only in 2012. In 2005, Makira, CAZ, COFAV and others, got temporary status. Temporary status is not a status

like a national park, nature reserve, or strict nature reserve. It is kind of a transition status (i.e., in the process of becoming established PAs). To our knowledge, Makira is the only one which has made it to the national park status, so far. We decided to avoid combining PAs that are fully established (MNP PAs) and others that are not yet fully established (NGO managed). (2) The management structure of the NGO managed PAs is quite complicated. They are not really managed by NGOs. They are co-managed. The NGOs are supporting organizations. In some cases, some portions are managed by the local communities. In other cases, the entire PAs are managed by the local communities. The portions managed by local communities are included in our CFM data. When the entire PAs are managed by local communities, they are in our analyses but as CFM not MNP.

Reviewer: • Do the authors have any insights from their research on what could be done to prevent a similar spike in deforestation after the crisis? I.e., given the urgency of the issue, offering more explicit recommendations for practitioners and future researchers would be appropriate.

- Response: thank you for this suggestion. We included recommendations for practitioners in the Discussion, last paragraph: “Lessons from these and similar studies suggest potential pathways for improving CFM performance during and after crises, such as by strengthening social cohesion, reinforcing local tenure and use rights, and increasing levels of support from government or non-governmental conservation organizations.”

Reviewer #3 (Remarks to the Author):

[Editor’s note: Reviewer 2 co-reviewed the manuscript with one of the reviewers who provided the listed reports as part of the Nature Communications initiative to facilitate training in peer review and appropriate recognition for co-reviewers]

- Response: thank you to all reviewers, we feel the comments have helped us strengthen the manuscript. We fully support training in peer review and appropriate recognition for reviewers!

References

1. Freyaldenhoven, S., Hansen, C. & Shapiro, J. M. Pre-Event Trends in the Panel Event-Study Design. *American Economic Review* **109**, 3307–3338 (2019).

2. Jackson, C. K., Johnson, R. C. & Persico, C. The Effects of School Spending on Educational and Economic Outcomes: Evidence from School Finance Reforms *. *The Quarterly Journal of Economics* **131**, 157–218 (2016).
3. Guryan, J. Desegregation and Black Dropout Rates. *American Economic Review* **94**, 919–943 (2004).
4. Miller, S., Johnson, N. & Wherry, L. R. Medicaid and Mortality: New Evidence From Linked Survey and Administrative Data*. *The Quarterly Journal of Economics* **136**, 1783–1829 (2021).
5. Dobkin, C., Finkelstein, A., Kluender, R. & Notowidigdo, M. J. The Economic Consequences of Hospital Admissions. *American Economic Review* **108**, 308–352 (2018).
6. Stuart, E. A. Matching methods for causal inference: A review and a look forward. *Stat Sci* **25**, 1–21 (2010).
7. Goodman-Bacon, A. Difference-in-differences with variation in treatment timing. *Journal of Econometrics* **225**, 254–277 (2021).
8. Miller, D. L. An Introductory Guide to Event Study Models. *Journal of Economic Perspectives* **37**, 203–230 (2023).
9. Angrist, J. D. & Pischke, J.-S. *Mastering 'Metrics: The Path from Cause to Effect*. (Princeton University Press, 2014).
10. Cunningham, S. *Causal inference: the mixtape*. (Yale University Press, 2021).
11. Blackman, A. & Villalobos, L. Use Forests or Lose Them? Regulated Timber Extraction and Tree Cover Loss in Mexico. *Journal of the Association of Environmental and Resource Economists* **8**, 125–163 (2021).

12. Solis, D., Cronkleton, P., Sills, E. O., Rodriguez-Ward, D. & Duchelle, A. E. Evaluating the Impact of REDD+ Interventions on Household Forest Revenue in Peru. *Frontiers in Forests and Global Change* **4**, (2021).
13. Jones, K. W. *et al.* Measuring the net benefits of payments for hydrological services programs in Mexico. *Ecological Economics* **175**, 106666 (2020).
14. Sharma, B. P. *et al.* Making incremental progress: impacts of a REDD+ pilot initiative in Nepal. *Environ. Res. Lett.* **15**, 105004 (2020).
15. Jones, K. W., Etchart, N., Holland, M., Naughton-Treves, L. & Arriagada, R. The impact of paying for forest conservation on perceived tenure security in Ecuador. *CONSERVATION LETTERS* **13**, (2020).
16. Ruggiero, P. G. C., Metzger, J. P., Reverberi Tambosi, L. & Nichols, E. Payment for ecosystem services programs in the Brazilian Atlantic Forest: Effective but not enough. *Land Use Policy* **82**, 283–291 (2019).
17. Arriagada, R., Villaseñor, A., Rubiano, E., Cotacachi, D. & Morrison, J. Analysing the impacts of PES programmes beyond economic rationale: Perceptions of ecosystem services provision associated to the Mexican case. *Ecosystem Services* **29**, 116–127 (2018).
18. Jones, K. W., Muñoz Brenes, C. L., Shinbrot, X. A., López-Báez, W. & Rivera-Castañeda, A. The influence of cash and technical assistance on household-level outcomes in payments for hydrological services programs in Chiapas, Mexico. *Ecosystem Services* **31**, 208–218 (2018).
19. Beauchamp, E., Clements, T. & Milner-Gulland, E. J. Assessing Medium-term Impacts of Conservation Interventions on Local Livelihoods in Northern Cambodia. *World Development* **101**, 202–218 (2018).

20. Jagger, P. & Rana, P. Using publicly available social and spatial data to evaluate progress on REDD+ social safeguards in Indonesia. *Environmental Science & Policy* **76**, 59–69 (2017).
21. Chervier, C. & Costedoat, S. Heterogeneous Impact of a Collective Payment for Environmental Services Scheme on Reducing Deforestation in Cambodia. *World Development* **98**, 148–159 (2017).
22. Rasolofoson, R. A., Ferraro, P. J., Jenkins, C. N. & Jones, J. P. G. Effectiveness of Community Forest Management at reducing deforestation in Madagascar. *Biological Conservation* **184**, 271–277 (2015).
23. Miteva, D. A., Murray, B. C. & Pattanayak, S. K. Do protected areas reduce blue carbon emissions? A quasi-experimental evaluation of mangroves in Indonesia. *Ecological Economics* **119**, 127–135 (2015).
24. Clements, T. & Milner-Gulland, E. J. Impact of payments for environmental services and protected areas on local livelihoods and forest conservation in northern Cambodia. *Conservation Biology* **29**, 78–87 (2015).
25. Costedoat, S. *et al.* How Effective Are Biodiversity Conservation Payments in Mexico? *PLOS ONE* **10**, e0119881 (2015).
26. Bauch, S. C., Sills, E. O. & Pattanayak, S. K. Have We Managed to Integrate Conservation and Development? ICDP Impacts in the Brazilian Amazon. *World Development* **64**, S135–S148 (2014).
27. Simonet, G., Subervie, J., Ezzine-de-Blas, D., Cromberg, M. & Duchelle, A. E. Effectiveness of a REDD+ Project in Reducing Deforestation in the Brazilian Amazon. *American Journal of Agricultural Economics* **101**, 211–229 (2018).

28. Ferraro, P. J. & Simorangkir, R. Conditional cash transfers to alleviate poverty also reduced deforestation in Indonesia. *Science Advances* **6**, eaaz1298 (2020).
29. Jones, K. W. & Lewis, D. J. Estimating the Counterfactual Impact of Conservation Programs on Land Cover Outcomes: The Role of Matching and Panel Regression Techniques. *PLOS ONE* **10**, (2015).
30. Ribas, L. G. S., Pressey, R. L. & Bini, L. M. Estimating counterfactuals for evaluation of ecological and conservation impact: an introduction to matching methods. *Biol Rev* brv.12697 (2021) doi:10.1111/brv.12697.
31. Hansen, M. C. *et al.* High-Resolution Global Maps of 21st-Century Forest Cover Change. *Science* **342**, 850–853 (2013).
32. Vieilledent, G. *et al.* Combining global tree cover loss data with historical national forest cover maps to look at six decades of deforestation and forest fragmentation in Madagascar. *Biological Conservation* **222**, 189–197 (2018).
33. Schleicher, J. *et al.* Statistical matching for conservation science. *CONSERVATION BIOLOGY* **34**, 538–549 (2020).
34. Ramaharitra Tondrasoa, T. Human dimension of conservation planning: the case of Madagascar at national and regional scales. (UC Berkeley, 2012).
35. SRTM. *Madagascar Digital Elevation Map*. <https://energydata.info/dataset/madagascar-elevation-2008/resource/fa701748-0cc9-452e-b126-ec16abae5512> (2000).
36. Fick, S. E. & Hijmans, R. J. WorldClim 2: new 1-km spatial resolution climate surfaces for global land areas. *International Journal of Climatology* **37**, 4302–4315 (2017).
37. WorldPop. *Global High Resolution Population Denominators Project*. <https://dx.doi.org/10.5258/SOTON/WP00674> (2018).

38. Diamond, A. & Sekhon, J. S. Genetic Matching for Estimating Causal Effects: A General Multivariate Matching Method for Achieving Balance in Observational Studies. *The Review of Economics and Statistics* **95**, 932–945 (2012).
39. Ho, D. E., Imai, K., King, G. & Stuart, E. A. MatchIt: Nonparametric Preprocessing for Parametric Causal Inference. *Journal of Statistical Software* **42**, 1–28 (2011).
40. Greifer, N. cobalt: Covariate Balance Tables and Plots. R package. (2022).
41. Busch, J. & Ferretti-Gallon, K. What Drives Deforestation and What Stops It? A Meta-Analysis. *Review of Environmental Economics and Policy* **11**, 3–23 (2017).
42. Barrett, C. B. Stochastic food prices and slash-and-burn agriculture. *Environment and Development Economics* **4**, 161–176 (1999).
43. Eklund, J. *et al.* Elevated fires during COVID-19 lockdown and the vulnerability of protected areas. *Nat Sustain* 1–7 (2022) doi:10.1038/s41893-022-00884-x.
44. Bergé, L., Krantz, S. & McDermott, G. fixest: Fast Fixed-Effects Estimations. (2023).
45. Harper, G. J., Steininger, M. K., Tucker, C. J., Juhn, D. & Hawkins, F. Fifty years of deforestation and forest fragmentation in Madagascar. *Environmental Conservation* **34**, 325–333 (2008).
46. Schwitzer, C. *et al.* Averting Lemur Extinctions amid Madagascar’s Political Crisis. *Science* **343**, 842–843 (2014).
47. Gardner, C. J. *et al.* The rapid expansion of Madagascar’s protected area system. *Biological Conservation* **220**, 29–36 (2018).
48. Robinson, B. E., Holland, M. B. & Naughton-Treves, L. Does secure land tenure save forests? A meta-analysis of the relationship between land tenure and tropical deforestation. *Global Environmental Change* **29**, 281–293 (2014).

49. Rakotonarivo, O. S. *et al.* Resolving land tenure security is essential to deliver forest restoration. *Commun Earth Environ* **4**, 1–8 (2023).
50. Montagne, P., Maafaka, R., Aubert, S., Andriambolanoro, D. & Randrianarivelo, G. La Sécurisation Foncière Relative dans le contexte de réforme foncière à Madagascar : le cas du kijana de Berinrinina. (2009).
51. Hajjar, R. *et al.* A global analysis of the social and environmental outcomes of community forests. *Nature Sustainability* **4**, 216–224 (2021).
52. Suzzi-Simmons, A. Status of deforestation of Madagascar. *Global Ecology and Conservation* **42**, e02389 (2023).
53. Karna, B. K., Shivakoti, G. P. & Webb, E. L. Resilience of community forestry under conditions of armed conflict in Nepal. *Environmental Conservation* **37**, 201–209 (2010).
54. Liévano-Latorre, L. F., Brum, F. T. & Loyola, R. How effective have been guerrilla occupation and protected areas in avoiding deforestation in Colombia? *Biological Conservation* **253**, 108916 (2021).
55. Burgess, R., Miguel, E. & Stanton, C. War and deforestation in Sierra Leone. *Environ. Res. Lett.* **10**, 095014 (2015).
56. Arakwiye, B., Rogan, J. & Eastman, J. R. Thirty years of forest-cover change in Western Rwanda during periods of wars and environmental policy shifts. *Reg Environ Change* **21**, 27 (2021).
57. Ordway, E. M. Political shifts and changing forests: Effects of armed conflict on forest conservation in Rwanda. *Global Ecology and Conservation* **3**, 448–460 (2015).

58. Llopis, J. C. *et al.* Effects of protected area establishment and cash crop price dynamics on land use transitions 1990–2017 in north-eastern Madagascar. *Journal of Land Use Science* **14**, 52–80 (2019).

Reviewer #1 (Remarks to the Author):

The authors have generally met all my suggestions and made it clearly regarding the temporal control of covariables impacts on deforestation. The manuscript has substantial improvement, but I still see some incorrect expressions, e.g., two Fig S7 were included in Supp. Please proofread it carefully before publication.

Reviewer #2 (Remarks to the Author):

Here are our review comments regarding the manuscript revision:

First, kudos to the other reviewers who so closely examined the data sources and analytical design. The corresponding changes to the ms. have strengthened it.

With regard to our review, overall, the authors did a good job reworking the discussion to provide the necessary nuance that the original manuscript was missing. In particular, the section in the discussion suggesting other avenues of research that could help explain why deforestation increased during the crisis, and how this effected CFM, were insightful.

However, we think there is still some disconnect between the results and the discussion. First, in line 316-317, the authors conclude "results indicate that the type of designation and management a site receives is an important determinant of performance." The authors acknowledge however that there is no data available on what different CFMS are managed for, such as whether CFMs are managed for subsistence or commercial production, which we would expect to have significant impacts on deforestation. This should be mentioned and highlighted as an area of further research in the main body of the paper. Official designations for protected areas (strict protection, sustainable use, etc) are often not recognized by PA managers struggling to implement regulations with limited budget and political support (ref below). Again, here we're calling for caution in inferring causality.

Second, the authors should make it clear that the comparison between deforestation in CFM vs MNP does not represent an estimate of avoided deforestation, nor account for potential leakage. To make this claim the authors would need to construct a counterfactual of what would happen in the absence of any conservation model (either CFM or MNP). There are other counterfactual studies that use this approach that the authors could refer to such as Baragwanath and Bayi's (2020) (note we are not expecting the authors re-do their analysis, but they should highlight this as a limitation of their study).

Third, while we understand it is not possible to provide specific CFM and MNP specific deforestation rates due to the large number of CFMs, we still believe providing estimates, perhaps by regions, would enhance the ability of practioneers to use this research to improve their work. An event study alone, still controls for time invariant factors, providing important insights into geographical differences in deforestation hotspots throughout the country. I.e. to contribute to future research, particularly by Malagasy students/experts, descriptive data by CFM and MNP area should be included in an annex.

In the authors' response to our review, they wrote "In other cases, the entire PAs are managed by the local communities. The portions managed by local communities are included in our CFM data. When the entire PAs are managed by local communities, they are in our analyses but as CFM not MNP." This noteworthy information should be disclosed in the methods in case others want to replicate this study. Also, we assume many readers will approach this paper as a comparison of national-govt managed, versus community managed PA effectiveness. If the interest of this study is two compare these two conservation models, without looking at the nuances of management more broadly, then we do not think it is appropriate to classify these MNP as CFM. Is CFM not a legal designation? This needs clarification in the final version.

Finally, we would also recommend the authors check for spatial autocorrelation (Oldekop 2019; Negret et al. 2020). Statistical matching, while a robust method, treats each pixel as independent. It is unlikely that observations are independent of each other however because if there is deforestation in one observation, it likely impacts neighboring observations. This spatial dependency appears to be ignored by the authors, but it could impact estimates as other scholars demonstrate (Schleicher et al. 2017; Negret et al. 2020).

Minor revisions:

- We like how the author incorporated concerns about tenure security into the discussion. In Line 32, it would also be worth noting that tenure security is likely to be influenced by a political crisis given that the state may no longer back up land tenure claims.
- Authors mention that the confounding factors “influence both the likelihood that a site was designated as CFM versus MNP, and also influence deforestation outcomes.” Authors should add a sentence or two about how these factors could influence designation as CFM vs MNP and/or which would be most important. Remoteness? Or perhaps MNPs were designated due to the presence of endangered wildlife?

Reviewer #3 (Remarks to the Author):

I co-reviewed this manuscript with one of the reviewers who provided the listed reports as part of the Nature Communications initiative to facilitate training in peer review and appropriate recognition for co-reviewers.

Response to reviewers

Reviewer #1 (Remarks to the Author):

The authors have generally met all my suggestions and made it clearly regarding the temporal control of covariables impacts on deforestation. The manuscript has substantial improvement, but I still see some incorrect expressions, e.g., two Fig S7 were included in Supp. Please proofread it carefully before publication.

Response: We thank the reviewer. We have fixed the Fig S7 label in the Supplement (Fig S7 is a multi-panel figure split across two pages, Fig S7a-d and Fig S7e-h, the second part of the figure was incorrectly labeled, we have fixed this now.) We have also carefully proof-read the manuscript and supplement.

Reviewer #2 (Remarks to the Author):

Here are our review comments regarding the manuscript revision:

First, kudos to the other reviewers who so closely examined the data sources and analytical design. The corresponding changes to the ms. have strengthened it.

Response: Thank you, we agree and are glad the changes have improved the manuscript.

With regard to our review, overall, the authors did a good job reworking the discussion to provide the necessary nuance that the original manuscript was missing. In particular, the section in the discussion suggesting other avenues of research that could help explain why deforestation increased during the crisis, and how this effected CFM, were insightful.

However, we think there is still some disconnect between the results and the discussion. First, in line 316-317, the authors conclude “results indicate that the type of designation and management a site receives is an important determinant of performance.” The authors acknowledge however that there is no data available on what different CFMS are managed for, such as whether CFMs are managed for subsistence or commercial production, which we would expect to have significant impacts on deforestation. This should be mentioned and highlighted as an area of further research in the main body of the paper. Official designations for protected areas (strict protection, sustainable use, etc) are often not recognized by PA managers struggling to implement regulations with limited budget and political support (ref below). Again, here we’re calling for caution in inferring causality.

Response: Thank you, we agree. The issue raised here is caused by how we used the terms “designation” and “management”. The two terms are somehow conflated, and we apologize for that. We looked at *de jure* designation and NOT *de facto* management. Our results are therefore limited to causal effects of *de jure* designation. We modified

the section to clarify this limitation and include a suggestion for future research on *de facto* management, as indicated by the reviewer: “our results indicate that the type of designation ~~and management~~ a site receives is an important determinant of performance. Previous research from a subset of four sites indicates that CFM contracts that prohibited commercial use performed better than CFM that allowed such uses³⁹. For the vast majority of CFM, however, there is a lack of data on specific contract terms or their implementation. Furthermore, even at the best of times, protected area managers in Madagascar struggle to implement regulations due to limited budgets and lack of political support (Gardner et al. 2018). Our analysis therefore focuses on *de jure* designation, as currently there is no national-scale data on *de facto* management of CFM nor MNP. Future research on how CFM and MNP are managed in practice would expand our understanding of how they interact with shocks like political unrest to affect deforestation.”

Reviewer: Second, the authors should make it clear that the comparison between deforestation in CFM vs MNP does not represent an estimate of avoided deforestation, nor account for potential leakage. To make this claim the authors would need to construct a counterfactual of what would happen in the absence of any conservation model (either CFM or MNP). There are other counterfactual studies that use this approach that the authors could refer to such as Baragwanath and Bayi’s (2020) (note we are not expecting the authors re-do their analysis, but they should highlight this as a limitation of their study).

Response: Thank you, we agree. From the title, we make clear that we look at “**The effect of a political crisis** on performance of community forests and protected areas in Madagascar”, not the effects of CFM vs. MNP protected areas on deforestation. Furthermore, our study builds on previous impact evaluation studies from Madagascar which explored avoided deforestation of CFM relative to unprotected forest (Rasolofoson et al. 2015), and avoided deforestation of protected areas relative to unprotected forest (Eklund et al. 2016, Llopis et al. 2019). These previous studies, and their conclusions, are summarized in the Introduction: “From 1990 to 2010, Madagascar’s protected areas were found to be effective at reducing deforestation, on average, but performance varied across time and space³⁴. In northeastern Madagascar, for example, the establishment of new protected areas initially exacerbated ongoing deforestation, but later reduced forest loss³⁵.” And: “Previous research found that CFM had no detectable impact on deforestation, on average, between 2000 and 2010, but contracts that prohibited commercial use of forest products did reduce deforestation³⁹. In the present analysis, we seek to build on these previous studies, with a new and more narrowly focused research question related to the effect of a political crisis on relative performance of CFM and MNP.

That said, comparing our results to “never protected” forest was something we considered carefully when planning this analysis. We ultimately decided against doing so. We have added some language explaining our reasoning in the Methods (“Statistical Matching” sub-section): “We considered comparing CFM and MNP performance to

“unprotected” forest. However, by the time of the crisis and the post-crisis period, there was very little forest that wasn’t under some kind of designation, due to the expansion of Madagascar’s protected area and CFM network after 2005 (Fig S1). Further, the forest that remained unprotected would not serve as good matches for CFM or MNP forest due to differences in location and other characteristics.”

Fig. S1. Community Forest Management areas (CFM), protected areas administered by Madagascar National Parks (MNP), and other System of Protected Areas (SAPM) Community Forest Management areas (CFM) (red), protected areas administered by Madagascar National Parks (MNP) (blue), and protected areas administered by other agencies (yellow). Only CFM and MNP established before 2005 (red with black hatching, blue with blue hatching) were included in the analysis. Forest cover 2020 (dark gray).

Reviewer: Third, while we understand it is not possible to provide specific CFM and MNP specific deforestation rates due to the large number of CFMs, we still believe providing estimates, perhaps by regions, would enhance the ability of practitioners to use this research to improve their work. An event study alone, still controls for time invariant factors, providing important insights into geographical differences in deforestation hotspots throughout the country. I.e. to contribute to future research, particularly by Malagasy students/experts, descriptive data by CFM and MNP area should be included in an annex.

Response: Thank you, and we understand the reviewer's interest in descriptive data. We provide national-level statistics on deforestation within CFM, MNP, and other forests in Table S1. After discussing this with our co-authors and with local experts, we feel strongly that providing such descriptive data by CFM and MNP site (or clusters of sites within the same region) is not desirable, as it would highlight individual sites (or regions) that experienced high levels of deforestation, without controlling for the fact that some sites/regions face significantly higher background levels of pressures (e.g. sites or regions with more dense road networks or larger population centers). The motivation of our methodological approach was to avoid this kind of naïve comparison. Furthermore, such information could be sensitive for our partners in Madagascar, who are working to protect forests and biodiversity with limited resources and under extremely challenging circumstances. It would be unfair and misleading to, for example, draw negative attention to certain sites/regions experiencing relatively high levels of deforestation without accounting for differences in location and other characteristics which may be driving forest loss.

Ideally, we would be able to evaluate the performance of individual sites while controlling for these confounding factors. Unfortunately, this is not feasible, as it's not possible to find good matches at the site level. For example, there were only 45 MNPs established prior to 2005 but more than 300 CFM sites. MNP are much larger in size and are located in more remote, higher elevation areas with steeper slopes and lower suitability for agriculture. This is why we relied on forest grid cells as our unit of analysis. We have added some language in the Methods to explain: "Due to their difference in number, size, and other characteristics, it is impossible to find good site-level matches for individual CFM or MNP sites, thus we focus on 90 m (and 270 m) forest grid cells as the unit of analysis."

Reviewer: In the authors' response to our review, they wrote "In other cases, the entire PAs are managed by the local communities. The portions managed by local communities are included in our CFM data. When the entire PAs are managed by local communities, they are in our analyses but as CFM not MNP." This noteworthy information should be disclosed in the methods in case others want to replicate this study. Also, we assume many readers will approach this paper as a comparison of national-govt managed, versus community managed PA effectiveness. If the interest of this study is two compare these two conservation models, without looking at the nuances of management more broadly, then we do not think it is appropriate to classify

these MNP as CFM. Is CFM not a legal designation? This needs clarification in the final version.

Response: We apologize for the confusion. CFM is indeed a legal designation. We should clarify the term “MNP” is not synonymous with all protected areas in Madagascar. MNP is a para-statal organization mandated by the government of Madagascar to manage many of Madagascar’s protected areas, specifically old protected areas established before 2005. We refer to “protected areas administered by MNP (MNP)” and focus on these in our analyses. There are however other protected areas not administered by MNP (most new protected areas established after 2005). Most of these new protected areas are co-managed between NGOs and local communities. A few are entirely managed by local communities. The portions managed by local communities are included in our analysis as CFM. The study therefore is still “a comparison of national-govt managed (MNP protected areas), versus community managed PA effectiveness”. We have added the following text: “Our analysis was somewhat complicated by issues of overlapping designation. In some cases, sites that were designated as protected areas by the government are partially or entirely managed by local communities. The portions managed by local communities are included in our analysis as CFM. Overlapping areas where designation of the site is unclear were excluded from our analysis (see for example Fig S4-S5).”

Gardner et al. 2018 (10.1016/j.biocon.2018.02.011) further explain:

Prior to 2003 all PAs in Madagascar were governed by the State through the parastatal ANGAP/MNP (though in some cases management was delegated to NGOs), but the Durban Vision saw the rewriting of the Protected Area Code to permit actors other than MNP to manage PAs within SAPM. All non-MNP PAs have a legally-recognized promoter, typically international or Malagasy NGOs (although also universities, mining companies and private individuals), but are generally governed in shared governance arrangements incorporating regional authorities and local communities (Alvarado et al., 2015; Virah-Sawmy et al., 2014). These governance structures have evolved iteratively: initial management plans of many sites proposed community management with promoter NGOs limited to a supporting role (e.g. Gardner et al., 2008), however this concealed the reality of promoters as de facto (co)managers, providing funds, technical capacity, direction and drive (Franks and Booker, 2015). In response, promoters must now be named as delegated managers of new PAs with responsibility for management to the State.

Most non-MNP PAs have multi-tiered governance structures incorporating i) an executive body/platform comprising the promoter and a community-based management committee, and ii) an orientation committee grouping regional authorities, relevant ministries and private sector representatives (e.g. tourism operators) (Franks and Booker, 2015; Virah-Sawmy et al., 2014). Depending on their size, the community-based management committees may be based around spatially nested hierarchies with two or three tiers: local management units (LMUs) are responsible for their own territories but elect representatives to sit on a federation of LMUs covering a larger area, and this in turn may elect

representatives to a central committee responsible for the whole protected area (Andriamalala and Gardner, 2010; Virah-Sawmy et al., 2014) (Fig. 2). In some PAs the LMUs are composed of management transfers enacted under CBNRM legislation and thus have a legal standing beyond that of the PA. In all cases these structures remain ‘works in progress’, and will require years of further experimentation and evolution before they are optimized.

Reviewer: Finally, we would also recommend the authors check for spatial autocorrelation (Oldekop 2019; Negret et al. 2020). Statistical matching, while a robust method, treats each pixel as independent. It is unlikely that observations are independent of each other however because if there is deforestation in one observation, it likely impacts neighboring observations. This spatial dependency appears to be ignored by the authors, but it could impact estimates as other scholars demonstrate (Schleicher et al. 2017; Negret et al. 2020).

Response: Thank you for raising this issue. We are aware of the issue of spatial autocorrelation and considered it in our study design. As described in the Methods, we clustered standard errors at the site level (where each site is a unique CFM or MNP.) This step accounts for unobservable variables common to observations within the same site, which will take care of part of any spatial autocorrelation. We added one sentence to the Methods to clarify this: “This step also addresses spatial autocorrelation between observations within the same site (Negret et al. 2020).”

To address the reviewer’s comment we have now applied multilevel clustering of standard errors. We clustered standard errors at the site level as described above, as well as at the region level (where regions were defined by the 22 administrative regions of Madagascar). This addresses potential lack of independence (i.e., autocorrelation) of observations within the same site as well as across different sites within the same region. The multilevel clustering did not affect our estimated effects (point estimates) but rendered the observed differences in the years 2014-2017 marginally significant (i.e., at 10% significance level, instead of at 5%). We have added a description of this new test of multilevel clustering to the Methods section: “We also tested multilevel clustering of standard errors at the site and region level (22 administrative regions). Multilevel clustering did not affect our point estimates but rendered the observed differences in the years 2014-2017 marginally significant ($p < 0.1$ instead of $p < 0.05$).”

Reviewer:

Minor revisions:

- We like how the author incorporated concerns about tenure security into the discussion. In Line 32, it would also be worth noting that tenure security is likely to be influenced by a political crisis given that the state may no longer back up land tenure claims.

Response: Thank you. In the Discussion, we added the suggested language: “Tenure security is likely to be impacted by a political crisis, however, given that the state may no longer enforce land tenure claims.”

Reviewer: Authors mention that the confounding factors “influence both the likelihood that a site was designated as CFM versus MNP, and also influence deforestation outcomes.” Authors should add a sentence or two about how these factors could influence designation as CFM vs MNP and/or which would be most important. Remoteness? Or perhaps MNPs were designated due to the presence of endangered wildlife?

Response: Thank you. We added the following language: “Because sites administered by MNP include the oldest protected areas in Madagascar, and were established primarily to shield biodiversity from human pressure, they were typically located in more remote, higher elevation areas, with fewer competing land uses, and are larger and more contiguous (Sussman et al. 1994). CFM are a newer designation and are intended for multiple use. CFM sites therefore tend to be smaller and established in areas with higher human pressure and closer to human settlements (Rasolofoson et al. 2015).”